# Insight into the evolution of microbial metabolism from the deep-branching bacterium, *Thermovibrio ammonificans*

Donato Giovannelli[1,2,3,4]*, Stefan M Sievert[5], Michael Hügler[6], Stephanie Markert[7], Dörte Becher[8], Thomas Schweder[7], Costantino Vetriani[1,9]*

[1]Institute of Earth, Ocean and Atmospheric Sciences, Rutgers University, New Brunswick, United States; [2]Institute of Marine Science, National Research Council of Italy, Ancona, Italy; [3]Program in Interdisciplinary Studies, Institute for Advanced Studies, Princeton, United States; [4]Earth-Life Science Institute, Tokyo Institute of Technology, Tokyo, Japan; [5]Biology Department, Woods Hole Oceanographic Institution, Woods Hole, United States; [6]DVGW-Technologiezentrum Wasser, Karlsruhe, Germany; [7]Pharmaceutical Biotechnology, Institute of Pharmacy, Institute of Pharmacy, Ernst-Moritz-Arndt-University Greifswald, Greifswald, Germany; [8]Institute for Microbiology, Ernst-Moritz-Arndt-University Greifswald, Greifswald, Germany; [9]Department of Biochemistry and Microbiology, Rutgers University, New Brunswick, United States

**Abstract** Anaerobic thermophiles inhabit relic environments that resemble the early Earth. However, the lineage of these modern organisms co-evolved with our planet. Hence, these organisms carry both ancestral and acquired genes and serve as models to reconstruct early metabolism. Based on comparative genomic and proteomic analyses, we identified two distinct groups of genes in *Thermovibrio ammonificans*: the first codes for enzymes that do not require oxygen and use substrates of geothermal origin; the second appears to be a more recent acquisition, and may reflect adaptations to cope with the rise of oxygen on Earth. We propose that the ancestor of the *Aquificae* was originally a hydrogen oxidizing, sulfur reducing bacterium that used a hybrid pathway for $CO_2$ fixation. With the gradual rise of oxygen in the atmosphere, more efficient terminal electron acceptors became available and this lineage acquired genes that increased its metabolic flexibility while retaining ancestral metabolic traits.

*For correspondence:
giovannelli@marine.rutgers.edu
(DG); vetriani@marine.rutgers.edu
(CV)

**Competing interests:** The authors declare that no competing interests exist.

## Introduction

Deep-branching, anaerobic, thermophilic *Bacteria* and *Archaea* inhabit relic environments that resemble the early Earth (*Baross and Hoffman, 1985*; *Martin et al., 2008*). Thermophily (*Di Giulio, 2003*, *2000*), anaerobic metabolism (*Baross and Hoffman, 1985*; *Martin et al., 2008*; *Schopf, 1983*) and reliance on substrates of geothermal origin are among the proposed ancestral traits of these microorganisms (*Baross and Hoffman, 1985*; *Di Giulio, 2003*, *2000*; *Lane et al., 2010*; *Martin et al., 2008*; *Russell and Martin, 2004*; *Schopf, 1983*). At the same time, their lineages have co-evolved with Earth and their genomes also carry more recently acquired traits. Therefore, these microorganisms can be used as models to reconstruct the evolution of metabolism.

*Thermovibrio ammonifican*s is part of the phylum *Aquificae*, a deep-branching group of thermophilic bacteria found in geothermal environments (*Lebedinsky et al., 2007*; *Sievert and Vetriani, 2012*). Based on phylogenetic analyses of the 16S rRNA gene as well as whole genomes, *Aquificae*

**eLife digest** Life may have arisen on our planet as far back as four billion years ago. Unlike today, the Earth's atmosphere at the time had no oxygen and an abundance of volcanic emissions including hydrogen, carbon dioxide and sulfur gases. These dramatic differences have led scientists to wonder: how did the ancient microorganisms that inhabited our early planet make a living? And how has microbial life co-evolved with the Earth?

One way to answer these questions is to study bacteria that live today in environments that resemble the early Earth. Deep-sea hydrothermal vents are regions of the deep ocean where active volcanic processes recreate primordial conditions. These habitats support microorganisms that are highly adapted to live off hydrogen, carbon dioxide and sulfur gases, and studying these modern-day microorganisms could give insights into the earliest life on Earth.

*Thermovibrio ammonificans* is a bacterium that was obtained from an underwater volcanic system in the East Pacific. Giovannelli et al. have now asked if *T. ammonificans* might have inherited some of its genetic traits from a long-gone ancestor that also thrived off volcanic gases. The genetic makeup of this microorganism was examined for genes that would help it thrive at a deep-sea hydrothermal vent. Next, Giovannelli et al. compared these genes to related copies in other species of bacteria to reconstruct how the metabolism of *T. ammonifican*s might have changed over time.

This approach identified a group of likely ancient genesthat allow a microorganism to use chemicals like hydrogen, carbon dioxide and sulfur to fuel its growth and metabolism. These findings support the hypothesis that an ancestor of *T. ammonifican*s could live off volcanic gases and that the core set of genes involved in those activities had been passed on, through the generations, to this modern-day microorganism. Giovannelli et al. also identified a second group of genes in *T. ammonificans* that indicate that this bacterium also co-evolved with Earth's changing conditions, in particular the rise in the concentration of oxygen.

The findings of Giovannelli et al. provide insight into how the metabolism of microbes has co-evolved with the Earth's changing conditions, and will allow others to formulate new hypotheses that can be tested in laboratory experiments.

are believed to be the earliest bacterial lineages having emerged on Earth along with the phyla *Thermotogae* and *Thermodesulfobacteria* (*Di Giulio, 2003*; *Pitulle et al., 1994*; *Battistuzzi et al., 2004*) (*Figure 1*, *Figure 1—figure supplement 1*). All the cultured members of this phylum are chemolithoautotrophs that use hydrogen as an energy source, have optimum growth temperatures between 65°C and 95°C and rely on the reductive tricarboxylic acid cycle (rTCA) to convert carbon dioxide into biomass (*Hügler et al., 2007*; *Hügler and Sievert, 2011*; *Sievert and Vetriani, 2012*) (*Table 1*), making them ideal candidates to investigate the evolution of early metabolism.

The ability to synthesize new biomass from inorganic precursors, *i.e.* autotrophic carbon fixation, is a critical step in the global carbon cycle and is considered a key invention during the early evolution of metabolism (*Braakman and Smith, 2012*; *Fuchs, 2011*; *Hügler and Sievert, 2011*). Among the six known pathways of carbon fixation (reviewed in *Fuchs, 2011*; *Hügler and Sievert, 2011*), the rTCA cycle (present in the *Aquificae*, among others) and the reductive acetyl-CoA pathway (present in acetogenic bacteria and methanogens), represent good candidates for the ancestral carbon fixation pathway (*Martin and Russell, 2003*; *Russell and Hall, 2006*). The emergence of a reductive acetyl-CoA pathway has been associated with the FeS-rich minerals at alkaline hydrothermal vents (*Martin and Russell, 2007*, *2003*; *Wächtershäuser, 1988a*) and the presence of homologous core enzymes in both *Bacteria* and *Archaea* potentially support its ancestral nature (*Fuchs, 2011*; *Russell and Martin, 2004*). The rTCA cycle is considered the modern version of a proposed prebiotic autocatalytic cycle fueled by the formation of the highly insoluble mineral pyrite in sulfur-rich hydrothermal environments (*Wächtershäuser, 1988b*, *1990*). The rTCA cycle is widespread among anaerobic and microaerophilic bacteria including all *Aquificae*, chemolithoautotrophic *Epsilonproteobacteria*, *Chlorobi*, and *Nitrospirae* (*Berg, 2011*; *Fuchs, 2011*; *Hügler and Sievert, 2011*) in addition to few other bacterial strains. A recent phylometabolic reconstruction hypothesized that all

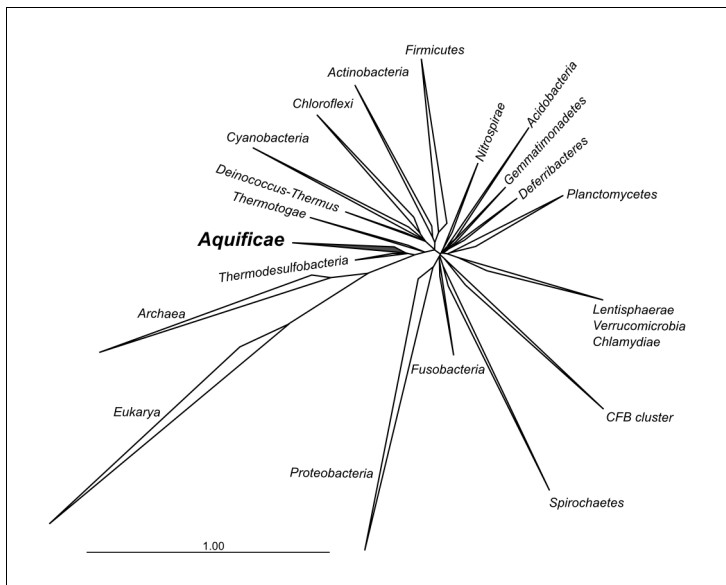

**Figure 1.** Phylogenetic tree of 16S rRNA sequences computed from the current version of the aligned 16S rRNA database obtained from the arb-SILVA project (http://www.arb-silva.de/).

The following figure supplement is available for figure 1:

**Figure supplement 1.** Maximum likelihood tree showing the 16S rRNA relationship of the *Aquificae* phylum.

extant pathways can be derived from an ancestral carbon fixation network consisting of a hybrid rTCA cycle / reductive acetyl-CoA pathway (*Braakman and Smith, 2012*).

Here, we used a comparative genomic approach coupled to proteomic analyses to reconstruct the central metabolism of *Thermovibrio ammonificans* (*Giovannelli et al., 2012*; *Vetriani et al., 2004*), and provide evidence of ancestral and acquired metabolic traits in the genome of this bacterium. We suggest that the last common ancestor within the *Aquificae* possessed the reductive acetyl-CoA pathway in addition to the previously described rTCA cycle. Furthermore, we show that these two pathways may still coexist within the *Desulfurobacteraceae* lineage. The simultaneous presence of both pathways of carbon fixation may represent a modern analog of the early carbon fixation phenotype, and suggests that the redundancy of central metabolic pathways could have been common in ancestral microorganisms.

## Results

### General genome structure and central metabolism of *T. ammonificans.*

The genome of *T. ammonificans* HB-1 consists of one chromosome and one plasmid of 1,682,965 bp and 76,561 bp, respectively (*Figure 2*). The genome encodes for 1890 genes, 1831 of which are protein coding genes and 75.16% of the protein coding genes could be assigned a putative function (*Giovannelli et al., 2012*). The chromosome contains three ribosomal RNA operons, the first two with a 5 S-23S-16S alignment (coordinates 149,693–152,859 bp and 1,168,776–1,172,141 bp antisense strand, respectively) and the other with a 16 S-23S-5S (coordinates 1,604,527–1,609,585 bp). The second copy of the operon is flanked by the largest CRISPR found in the genome (*Figure 2*, circle 6). Several other repeats were identified in the chromosome (*Figure 2*, circle 5). The plasmid contained mainly ORFs with unknown hypothetical functions (*Figure 2*, circle 9) with the exception of a putative RNA polymerase sigma factor of probable archaeal origin, a DNA Topoisomerase I (43.2% similarity to *Hydrogenivirga* sp. 128–5 R1-6), a type II secretion protein E (identified also in the preliminary proteome, *Figure 3*, *Figure 3—source data 1*) and an ArsR-like regulatory protein, both of

**Table 1.** Characteristic of representative members of *Aquificae* phylum.

| Family | Organism | Growth temp | Energy source | Carbon source | Terminal electron acceptor | Isolated from | Genome sequence accession number | References |
|---|---|---|---|---|---|---|---|---|
| *Aquificaceae* | *Aquifex aeolicus* VF5 | 95°C | $H_2$ | $CO_2$ | $O_2$ | Underwater volcanic vents, Aeolic Islands Sicily, Italy | NC_000918.1 | (*Deckert et al., 1998*; *Huber et al., 1992*) |
| | *Hydrogenivirga sp.*128–5 R1-1[1] | 75°C | $S_2O_3^{2-}$, $S_0$ | $CO_2$ | $NO_3^-$, $O_2$ | Lau Basin hydrothermal vent area, Pacific Ocean | NZ_ABHJ00000000 | (*Nakagawa et al., 2004*; *Reysenbach et al., 2009*) |
| | *Hydrogenobacter thermophilus* TK-6 | 75°C | $H_2$ | $CO_2$ | $O_2$ | Hot springs in Izu and Kyushu, Japan | NC_017161.1 | (*Arai et al., 2010*; *Kawasumi et al., 1984*) |
| | *Hydrogenobaculum sp.* Y04AAS1[*] | - | - | - | - | Marine hydrothermal area, Vulcano Island, Italy | NC_011126.1 | (*Reysenbach et al., 2009*; *Stohr et al., 2001*) |
| | *Thermocrinis ruber* DSM 12173 | 85°C | $H_2$, $S_2O_3^{2-}$, $S_0$ | $CO_2$ | $O_2$ | Octopus Spring, Yellowstone National Park, Wyoming, USA | PRJNA75073 | (*Huber et al., 1998*) |
| *Desulfurobacteriaceae* | *Balnearium lithotrophicum* 17S | 70–75°C | $H_2$ | $CO_2$ | $S_0$ | Deep-sea hydrothermal vent chimney, Suiyo Seamount, Japan | Not sequenced | (*Takai et al., 2003b*) |
| | *Desulfurobacterium thermolithotrophum* DSM 11699 | 70°C | $H_2$ | $CO_2$ | $S_0$, $S_2O_3^{2-}$, $SO_2^-$ | Deep-sea hydrothermal chimney, mid-Atlantic ridge, Atlantic Ocean | NC_015185.1 | (*Göker et al., 2011*; *L'Haridon et al., 1998*) |
| | *Phorcysia thermohydrogeniphila* HB-8 | 75°C | $H_2$ | $CO_2$ | $S_0$, $NO_3^-$ | Tube of *Alvinella pompejana* tubeworms, deep-sea hydrothermal vents 13 °N, East Pacific Rise, Pacific Ocean | Not sequenced | (*Pérez-Rodríguez et al., 2012*) |
| | *Thermovibrio ammonificans* HB-1 | 75°C | $H_2$ | $CO_2$ | $S_0$, $NO_3^-$ | Deep sea hydrothermal vents 9°N, East Pacific Rise, Pacific Ocean | NC_014926.1 | (*Giovannelli et al., 2012*; *Vetriani et al., 2004*) |
| *Hydrogenothermaceae* | *Hydrogenothermus marinus* VM1 | 65°C | $H_2$ | $CO_2$ | $O_2(1–2\%)$ | Deep sea hydrothermal vents 9°N, East Pacific Rise, Pacific Ocean | Not sequenced | (*Stohr et al., 2001*) |
| | *Persephonella marina* EX-H1 | 73°C | $S_0$ | $CO_2$ | $O_2$, $NO_3^-$ | Deep sea hydrothermal vents 9°N, East Pacific Rise, Pacific Ocean | NC_012440.1 | (*Götz et al., 2002*; *Reysenbach et al., 2009*) |
| | *Sulfurihydrogenibium sp.* YO3AOP1[1] | 70°C | $S_2O_3^{2-}$, $S_0$ | $CO_2$ | $O_2$ | Calcite Hot Springs, Yellowstone National Park, USA | NC_010730.1 | (*Reysenbach et al., 2009*; *Takai et al., 2003a*) |

*Table 1 continued*

| Family | Organism | Growth temp | Energy source | Carbon source | Terminal electron acceptor | Isolated from | Genome sequence accession number | References |
|---|---|---|---|---|---|---|---|---|
| | *Venenivibrio stagnispumantis* | 70°C | $H_2$ | $CO_2$ | $O_2$ | Terrestrial hot spring Champagne Pool, Waiotapu, New Zealand | Not sequenced | (*Hetzer et al., 2008*) |
| *Incertae sedis* | *Thermosulfidibacter takai* ABI70S6[T] | 70°C | $H_2$ | $CO_2$ | $S_0$ | deep-sea hydrothermal field at Southern Okinawa Trough, Japan | Not sequenced | (*Nunoura et al., 2008*) |

*− Strain not formally described whose genome sequence is available. For these strains, the physiological information reported in Table S1 have been collected from MIG associated with the sequencing or from the closest validly published species.

which had homologs among the *Firmicutes*. All other plasmid genes were ORFans with no significant similarities in the database and contained highly repeated DNA (*Figure 2*, circle 5).

By combining genome-scale analyses with known physiological traits, we were able to reconstruct the metabolic network of *T. ammonificans* (*Figure 3*). As part of the central metabolism of *T. ammonificans,* we identified the major pathways for carbon fixation, hydrogen oxidation and nitrate reduction to ammonia, and confirmed them by identifying the corresponding expressed enzymes in the proteome (*Figure 3*, *Figure 3—source data 1*). Further, we compared the central metabolism of *T. ammonificans* with closely related species within the *Aquificae,* and with ecologically similar *Epsilonproteobacteria.* Comparative genomic analyses with closely related strains provide information about the evolutionary history of the group, while comparison with phylogenetically distant, but ecologically similar species may reveal common adaptive strategies to similar habitats.

## The proteome of *T. ammonificans*

780 proteins, comprising approximately 43% of all predicted ORFs in the database, were identified in cell samples of *T. ammonificans* grown under nitrate reducing conditions (*Figure 3*, *Figure 3— source data 1*), indicating a very high congruence between database entries and sample peptides. Among the most abundant proteins were – besides those involved in general housekeeping functions like the translation elongation factor Tu, ribosomal proteins and chaperones – many rTCA cycle enzymes. The four key enzymes of the rTCA cycle (see next paragraph for details) together yielded a relative abundance of 6% of all identified proteins in the samples. Most of the enzymes putatively involved in the reductive acetyl-CoA pathway (*Table 2*), including the carbon-monoxide dehydrogenase (CodH), were also detected, although at a lower abundance of 1.25% of all proteins.

## Carbon fixation

The genome and proteome of *T. ammonificans* (*Figure 3*) supports previous evidence, based on detection of key genes and measurements of enzyme activities, that carbon fixation occurs via the rTCA cycle (*Hügler et al., 2007*). The rTCA is widespread among anaerobic and microaerophilic bacteria including all *Aquificae*, chemolithoautotrophic *Epsilonproteobacteria*, *Chlorobi*, *Nitrospina* aaaand *Nitrospirae,* in addition to few other bacterial strains (*e.g. Desulfobacter hydrogenophilus*, *Magnetococcus marinus* MC-1 and the endosymbiont of the vent tubeworm, *Riftia pachyptila*; *Hügler and Sievert, 2011*). We identified all the genes responsible for the functioning of the rTCA in the genome and proteome of *T. ammonificans* (*Figure 3*), including the key enzymes fumarate reductase (*frdAB*, gene loci Theam_1270–1275), ATP-citrate (pro-S)-lyase (*aclAB*, 1021–1022), 2-oxoglutarate synthase (*oorABCD*, 1410–1413) and isocitrate dehydrogenase (*idh2*, 1023). *T. ammonificans* synthesizes 2-oxoglutarate from succinyl-CoA rather than via citrate, a feature in common with methanogenic *Archaea* and *Desulfurococcales*, and present in numerous strict anaerobes that use the rTCA cycle (*Fuchs, 2011*). Comparative analyses showed that several genes associated with the rTCA cycle and the gluconeogenesis pathway were highly conserved in all *Aquificae*, with the

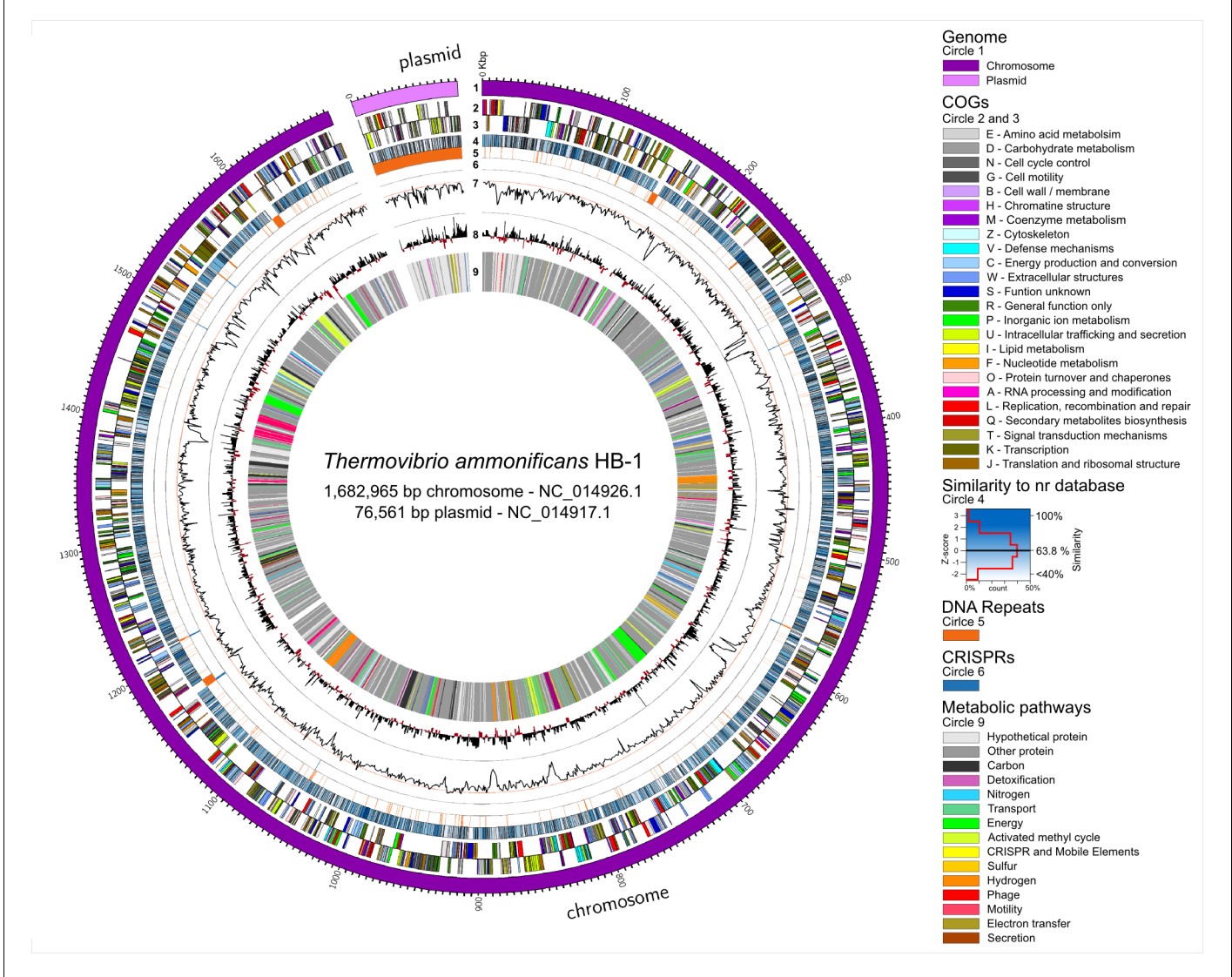

**Figure 2.** *Thermovibrio ammonificans* HB-1 genomic map highlighting main genomic features. The circular chromosome and plasmid are drawn together. Several other repeats were identified in the chromosome (circle 5). From the outer circle: 1 – Dimension of the genome in base pairs (chromosome in violet, plasmid in pink); 2 and 3 – sense and antisense coding sequences colored according to their COGs classification (see legend in the figure); 4 – similarity of each coding sequence with sequences in the non-redundant database; 5 – position of tandem repeat in the genome. A total of 46 tRNA were identified, comprising all of the basic 20 amino acid plus selenocysteine; 6 – position of CRISPRs; 7 – GC mol% content, the red line is the GC mean of the genome (52.1 mol%); 8 – GC skew calculated as G+C/A+T; 9 – localization of coding genes arbitrarily colored according to the metabolic pathways reconstructed (see *Figure 1*). Colors are consistent trough the entire paper. Total dimension and accession number for the chromosome and plasmid are given.

exception of the genes coding for the ATP-citrate lyase and the isocitrate-dehydrogenase, which are absent in the *Aquificaceae* (see details below) (*Hügler et al., 2007*). In general, *T. ammonificans* genes had a high percent identity to the genes of *D. thermolithotrophum,* its closest relative whose genome sequence is available.

Two separate variants of the rTCA cycle are known (*Braakman and Smith, 2012*; *Hügler et al., 2007*). In the two-step variant – also known as symmetric variant, with respect to the enzymatic reaction catalyzed in the two arcs of the rTCA cycle – citrate cleavage is accomplished by the combined action of the enzymes citryl-CoA synthetase (CCS) and citryl-CoA lyase (CCL) and the carboxylation of 2-oxoglutarate is catalyzed by 2-oxoglutarate carboxylase and oxalosuccinate reductase

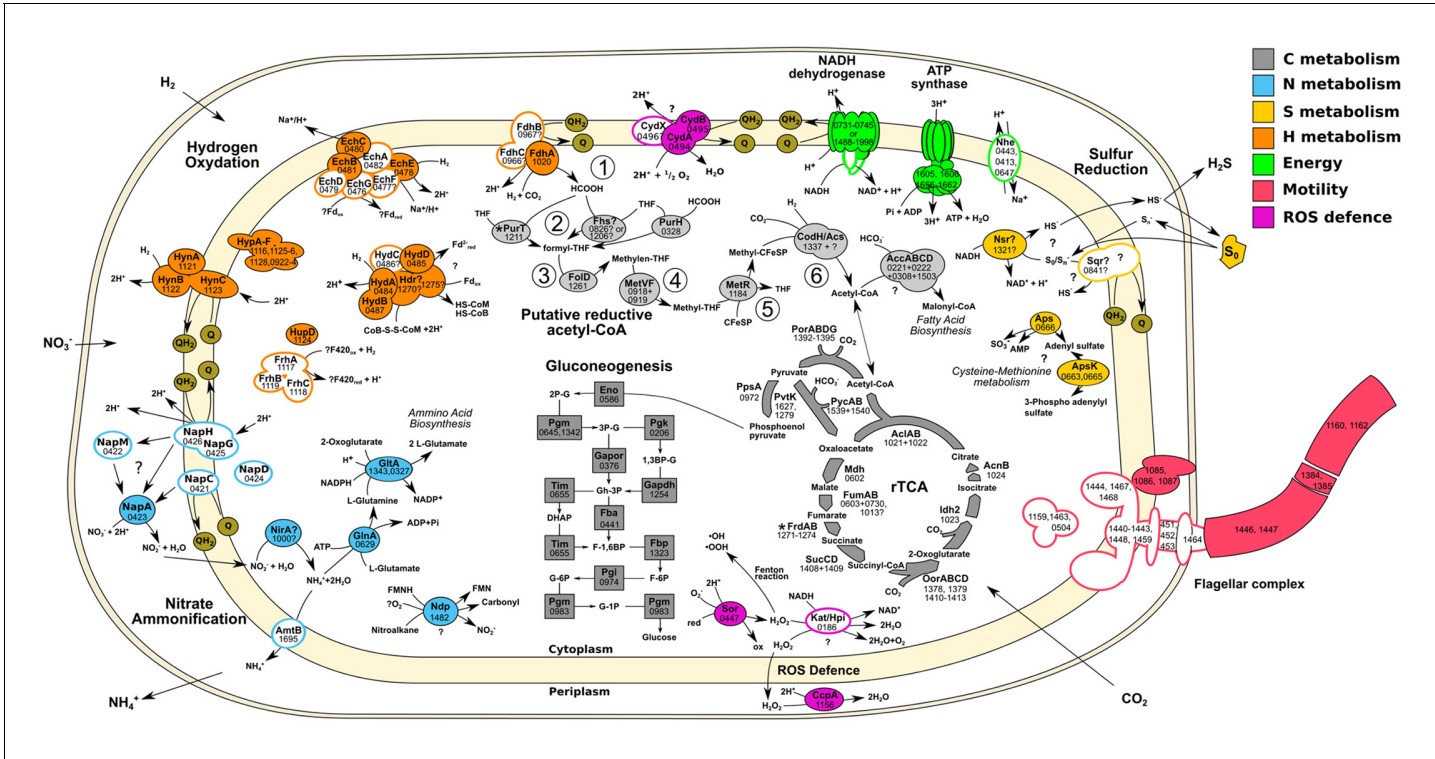

**Figure 3.** Central metabolism of *T. ammonificans* HB-1. Enzyme names are reported together with the gene locus number (Theam_*number*). Primary compounds involved in reactions were also reported, however visualized reactions are not complete. Pathways were arbitrarily color coded according to their reconstructed function and are consistent throughout the paper. Solid shapes represent genes for which the enzyme was found in the proteome, while outlined shapes were only identified in the genome. Circled numbers 1 to 6 represent reaction numbers of the putative reductive acetyl-CoA pathway as described in the text. Abbreviations: *NITRATE AMMONIFICATION*: NapCMADGH – Periplasmic nitrate reductase complex; NirA – Putative nitrite reductase; AmtB – ammonia transporter; GlnA – L-Glutamine synthetase; GltA – Glutamate synthetase. *HYDROGEN OXIDATION*: HynABC – Ni-Fe Membrane bound hydrogenase; FrhACB – Cytosolic Ni-Fe hydrogenase/putative coenzyme F420 hydrogenase; HupD – Cyrosolic Ni-Fe hydrogenase maturation protease; HypA-F – Hydrogenases expression/synthesis accessory proteins; EchABCEF – Ech membrane bound hydrogenase complex; HydAB – Cytosolic Ni-Fe hydrogenases potentially involved in ferredoxin reduction; Hdr? – Missing heterodisulfide reductase CoB-CoM; FdhABC – Formate dehydrogenase. *ENERGY PRODUCTION*: NADH dehydrogenase and ATP synthetase are reported without the names of the single units; Nhe – Sodium/hydrogen symporter. *SULFUR REDUCTION*: Sqr – Putative sulfate quinone reductase involved in sulfur respiration; Nsr? – FAD/NAD nucleotide-disulphide oxidoreductase; Aps – Sulfate adenylyl transferase and kinase involved in assimilation of sulfur. *FLAGELLAR COMPLEX*: for simplicity single unit names are not reported. *REDUCTIVE ACETYL-CoA*: Fhs – Proposed reverse formyl-THF deformylase; PurHT – Formyltransferases (phosphoribosylaminoimidazolecarboxamide and phosphoribosylglycinamide) FolD – Methenyl-THF cyclohydrolase and dehydrogenase; MetVF – Methylene-THF reductase; MetR – Putative methyl-transferase; CodH – Carbon monoxyde dehydrogenase; AcsA – Acetyl-CoA ligase/synthase; AccABCD – Acetyl-CoA carboxylase. *REDUCTIVE CITRIC ACID CYCLE*: AclAB – ATP-citrate lyase; Mdh – Malate dehydrogenase; FumAB – Fumarate hydratase; FrdAB – Fumarate reductase; SucCD – Succinyl-CoA synthetase; OorABCD – 2-Oxoglutarate synthase; Idh2 – Isocitrate dehydrogenase/2-oxoglutarate carboxylase; AcnB – Aconitate hydratase; PorABDG – Pyruvate synthase; PycAB - Pyruvate carboxylase; PpsA – Phosphoenolpyruvate synthase water dikinase; PyK – Pyruvate:water dikinase. *GLUCONEOGENESIS*: Eno – Enolase; Pgm – Phosphoglycerate mutase; Pgk – Phosphoglycerate kinase; Gapor – Glyceraldehyde-3-phosphate dehydrogenase; Gapdh – Glyceraldehyde 3-phosphate dehydrogenase; Fba – Predicted fructose-bisphosphate aldolase; Tim – Triosephosphate isomerase; Fbp – Fructose-1,6-bisphosphatase I; Pgi – Phosphoglucose isomerase; Pgm – Phosphoglucomutase.

The following source data and figure supplements are available for figure 3:

**Source data 1.** List of the proteins identified in the proteome of *T. ammonificans* grown under nitrate reducing conditions.

**Figure supplement 1.** Reductive TCA cycle variants found in extant bacterial lineages.

**Figure supplement 2.** The activated methyl cycle of *T. ammonificans* reconstructed from the genome.

**Table 2.** Enzymes involved in the putative reductive acetyl-CoA pathway in *T. ammonificans* HB-1, closest relative homolog and homologs within the Aquificae phylum.

| Reaction | Putative enzyme | Putative gene locus | Closest relative | Putative origin‡ | Desulfurobacteriaceae | | Hydrogenothermaceae | | Aquificaceae | | | | |
|---|---|---|---|---|---|---|---|---|---|---|---|---|---|
| | | | | | D. thermolithotrophum | Desulfurobacterium sp. TC5-1 | P. marina | Sulfurihydrogenibium sp. YO3AOP1 | H. thermophilus | Hydrogenobaculum sp. YO4AAS1H | Hydrogenivirga sp. 128-5 R1-6 | T. ruber | A. aeolicus |
| 1 | Formate dehydrogenase | Theam_1020 | Nitratiruptor sp. SB155-2 sim. 55% YP_001357016 | Methanogens | f.e.§ - 30% YP_00428203# | -** | 56% YP_2730364 | f.e.§ - 32% YP_001930236 | f.e.§ - 33% YP_003433330 | - | - | f.e.§ - 27% YP_003474076 | - |
| 2* | 5-formyltetrahydrofolate cyclo-ligase | Theam_1206 | Hydrogenivirga sp. 128-5 R1-1 sim. 41% WP_008285842 | Deltaproteobacteria / Gram + | 41% YP_004281339 | 44% WP_022846876.1 | f.e.§ - 36% YP_002729868 | f.e.§ - 38% YP_001931834 | f.e.§ - 38% YP_003431710 | f.e.§ - 30% YP_002121539 | 41% WP_008285842 | 40% YP_003472981 | 43% 1SOU_A |
| | formyltetrahydrofolate deformylase / hydrolase | Theam_0826 | Hydrogenivirga sp. 128-5 R1-1 sim. 70% WP_008287030 | Bacteria | 82% YP_004281077 | 77% WP_022846479 | - | - | - | - | 70% WP_008287030 | 65% NP_214247 | - |
| | phosphoribosylglycinamide formyltransferase 2 | Theam_1211 | P. marina sim. 76% YP_002731257 | Methanogens / Deltaproteobacteria | 78% YP_004281335 | 64% WP_022847512 | 76% YP_002731257 | 73% YP_001931730 | f.e.§ - 28% YP_003433072 | 67% YP_007499479 | 74% WP_008288476 | f.e.§ - 25% YP_003473050 | f.e.§ - 30% NP_213168 |
| | phosphoribosylaminoimidazole carboxamide formyltransferase /IMP cyclohydrolase | Theam_0328 | P. marina sim. 59% YP_002731268 | Aquificae / Bacteria | 86% YP_004281681 | 78% WP_022847359 | 59% YP_002731268 | 56% YP_001931239 | 53% YP_003433270 | 50% YP_007499473 | 57% WP_008288360 | 52% YP_003474337 | 56% NP_214344 |
| 3 | methylenetetrahydrofolate dehydrogenase / methenyltetrahydrofolate cyclohydrolase | Theam_1261 | A. aeolicus sim. 60% NP_214304 | Aquificae / Gram + | 87% YP_004281147 | 74% WP_022846217 | 55% YP_002731476 | 58% YP_001931240 | 59% YP_005511318 | 58% YP_001931240 | - | 58% YP_003473104 | 60% NP_214304 |
| 4 | methylenetetrahydrofolate reductase | Theam_0919 | Methanosarcina barkeri sim. 48% YP_305816 | Methanogens | 87% YP_004281147 | 74% WP_022846217 | 55% YP_002731476 | 58% YP_001931240 | 59% YP_005511318 | 58% YP_002121558 | f.e.§ - 24% WP_008286783 | 58% YP_003473104 | 60% NP_214304 |
| 5 | 5-methyltetrahydrofolate corrinoid/iron sulfurprotein methyltransferase | Theam_1184 | Clostridium glycolicum sim. 42% WP_018589788 | Gram + | 82% YP_004281484 | 72% WP_022846603 | f.e.§ - 31% YP_002729885 | f.e.§ - 38% YP_001930381 | f.e.§ - 37% YP_003432344 | f.e.§ - 34% YP_002122031 | f.e.§ - 38% WP_008287326 | f.e.§ - 36% YP_003473438 | f.e.§ - 34% NP_213375 |
| 6† | Carbon-monoxide dehydrogenase | Theam_1337 | Candidatus Methanoperedens sp. BLZ1 sim. 56% KPQ43483 | Archaea / Gram + | 87% WP_013638427 38% WP_013638030 | 76% WP_022847143 | 41% YP_002729904 | - | - | - | - | - | - |

* – Reactions are numbered according to **Figure 1**. For reaction two we reported the possible enzymes that could substitute the missing 10-formyl-THF synthetase (Fhs). The enzymes are listed in order of decreasing likelihood of their involvement in the reaction based on putative substrate affinity.

† – Reaction six is catalyzed by the Acs/CodH complex, reported here separately.

‡ – Putative origin of the gene was calculated as consensus taxonomic assignment of the first 100 bastp hits against the nr database. Double assignment implies equal number of assignments to the two taxonomic groups.

§ – f.e. = functional equivalent annotated in the genome with similarity below 40% with *T. ammonificans* equivalent gene. Not considered a true homolog in the present study.

# – Pairwise similarity to *T. ammonificans* translated gene and accession number of the homologs.

** – Missing homolog or functional annotated equivalent in the genome.

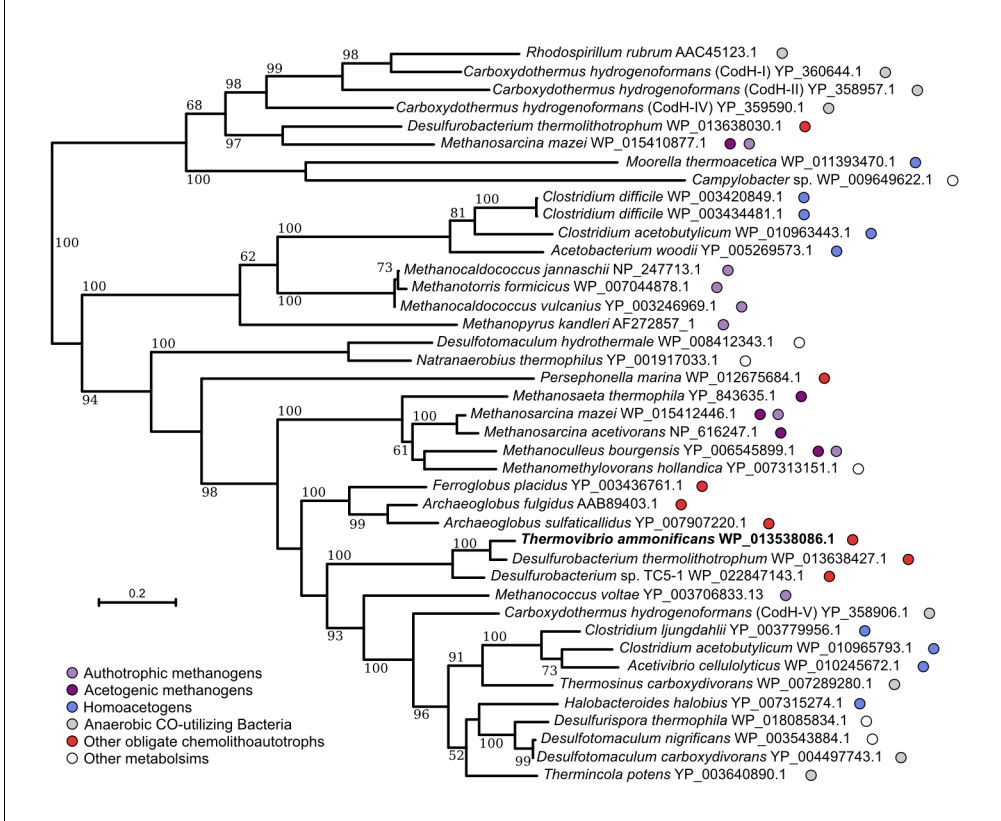

**Figure 4.** Neighbor-joining tree showing the position of the carbon monoxide dehydrogenase, CodH, of *T. ammonificans* (in bold). Bootstrap values based on 1000 replications are shown at branch nodes. Only bootstrap values above 50% are reported. Bar, 10% estimated substitutions. The metabolism of each organism is reported on the side.

(*Aoshima, 2007*; *Aoshima and Igarashi, 2008*; *Hügler and Sievert, 2011*). This variant of the rTCA cycle was previously proposed as ancestral (*Braakman and Smith, 2012*; *Hügler et al., 2007*). By contrast, in the one-step variant of the rTCA cycle, also known as asymmetric variant and more widely distributed, the cleavage of citrate and the carboxylation of 2-oxoglutarate are performed in a one-step fashion (*Braakman and Smith, 2012*; *Hügler et al., 2007*)(*Figure 3*, *Figure 3—figure supplement 1*). This variant is found in *T. ammonificans*.

The genome of *T. ammonificans* also codes for an incomplete reductive acetyl-CoA pathway (*Figure 3* and *Table 2*). The key enzyme of this carbon fixation pathway, which is found in acetogens, methanogens, sulfate-reducers and anaerobic ammonium oxidizers (anammox; *Berg et al., 2010*), is the bifunctional heterotetramer enzyme carbon monoxide dehydrogenase/acetyl-CoA synthase (CODH/ACS). The reductive acetyl-CoA pathway is believed to be the most ancient carbon fixation pathway on Earth (*Fuchs, 2011*). In *T. ammonificans*, the genes coding for the CO-dehydrogenase (*codh* catalytic subunit Theam_1337; *Table 2* and *Figure 3*) is present and also expressed. However, rather than coding for the classical bifunctional type II CodH, *T. ammonificans* codes for the type V, which is present in numerous deep-branching organisms (*Figure 4*) and whose function is unknown (*Techtmann et al., 2012*). Sequence analysis of the *T. ammonificans* CodH protein sequence revealed the presence of conserved residues necessary for the interaction with Acs and for the coordination of the NiFeS cluster. We hypothesize that the type V CodH, which, based on structural and size considerations may represent the most ancestral version of this class of enzymes (Frank Robb, personal communication), may catalyze the reduction of $CO_2$ to CO. The *codh* gene of *T. ammonificans* shared high similarity with that of *D. thermolithotrophum* (87 similarity), while the similarity with the next closest homologous genes dropped below 54%. The genome of *D. thermolithotrophum* also codes for a classical bifunctional Type II CodH (Dester_0417 and Dester_0418; *Figure 4*) in

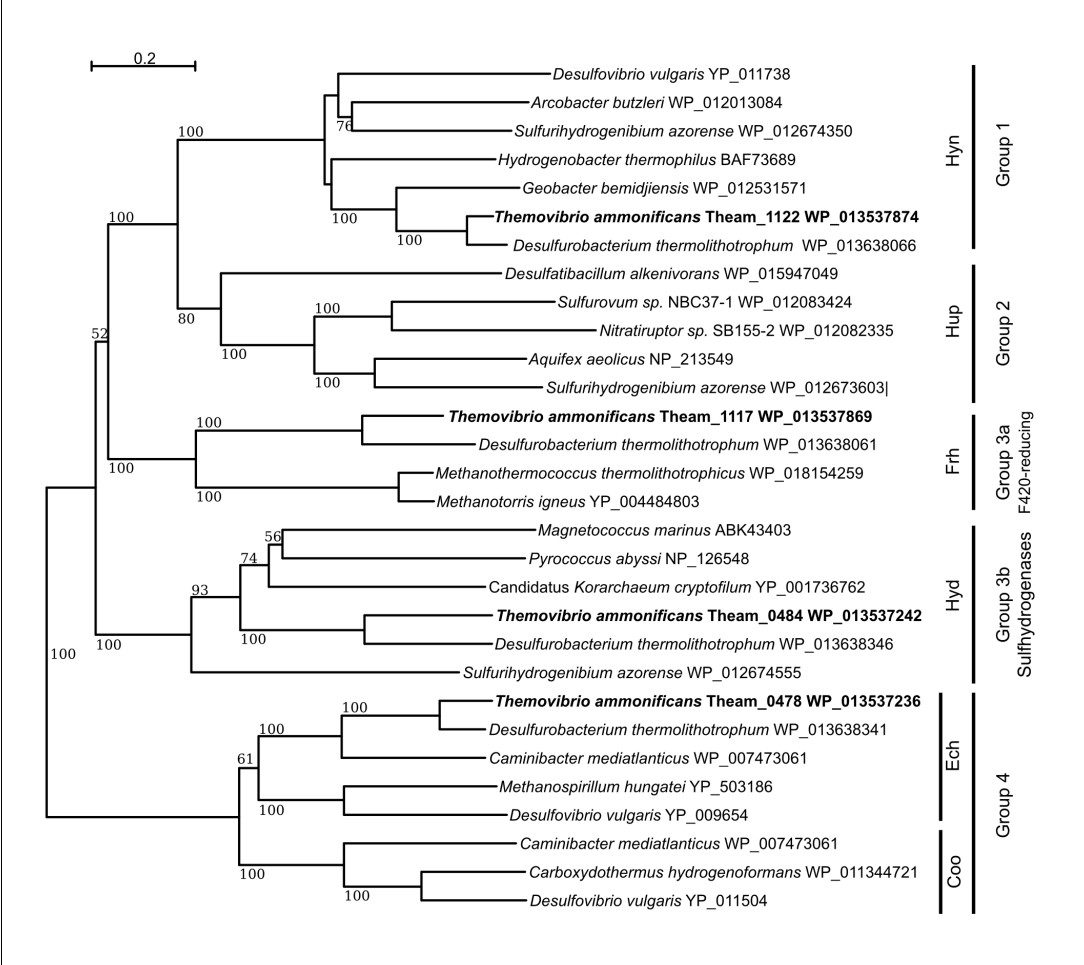

**Figure 5.** Neighbor-Joining tree showing the position and classification of the [NiFe]-hydrogenases found in the genome of *T.ammonificans*. Bootstrap values based on 1000 replication are shown at branch node. Bar, 20% estimated substitution.

addition to the type V CodH. The two genomes also share a CodH iron-sulfur accessory protein (Theam_0999 and Dester_0418, 81% similarity). Additional laboratory analyses will be required to confirm the role of the type V CodH in carbon fixation.

## Gluconeogenesis

*T. ammonificans* uses the Embden-Meyerhof-Parnas pathway to synthesize pentose and hexose monosaccharides. We identified in the genome the key genes of this pathway, fructose-1,6-biphosphatase (fbp, Theam_1323; identified also in the proteome, *Figure 3*) and all other genes necessary for the functioning of the pathway (*Figure 3*). While the *fbp* gene is present in *Desulfurobacterium thermolithotrophum*, and in the genome of *Sulfurohydrogenibium* sp. YO3AOP1, *Persephonella marina* and *Hydrogenivirga* sp. 128–5 R1-6, it is absent from the genome of *A. aeolicus* (*Deckert et al., 1998*) and *H. thermophilus*, suggesting that in those species an alternative unidentified pathway may be active.

## Hydrogen oxidation

Hydrogen is the only electron donor used by *T. ammonificans* and its importance in the metabolism of this bacterium is reflected by the diversity of hydrogenases found in the genome, whose encoding genes are organized in two large clusters (*Figure 2* circle 9). The first cluster is composed of eight genes encoding for the complete Group 4 Ech hydrogenases (*echABCDEFG*, Theam_0476–0482)

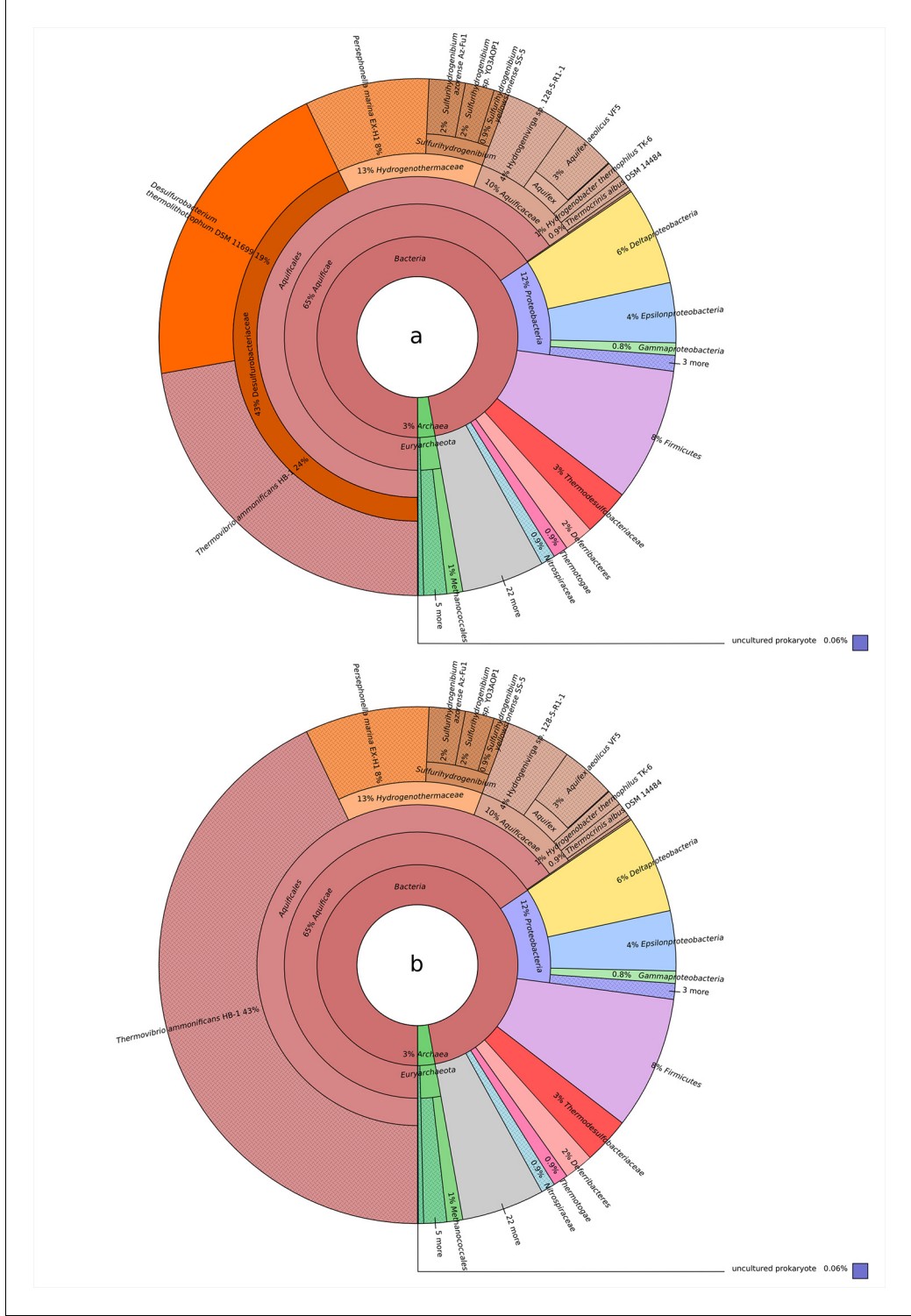

**Figure 6.** Blastp best hit (cut off at 40% similarity) for the *T. ammonificans* CDS: (a) *Desulfurobacterium thermolithotrophum* is included in the database; (b) *D. thermolithotrophum* and *Desulfurobacterium* sp. TC5-1 are excluded from the database (best hits are outside of the *Desulfurobacteriaceae* family). The interactive versions of the Krona plots are available for download at DOI: 10.6084/m9.figshare.3178528.

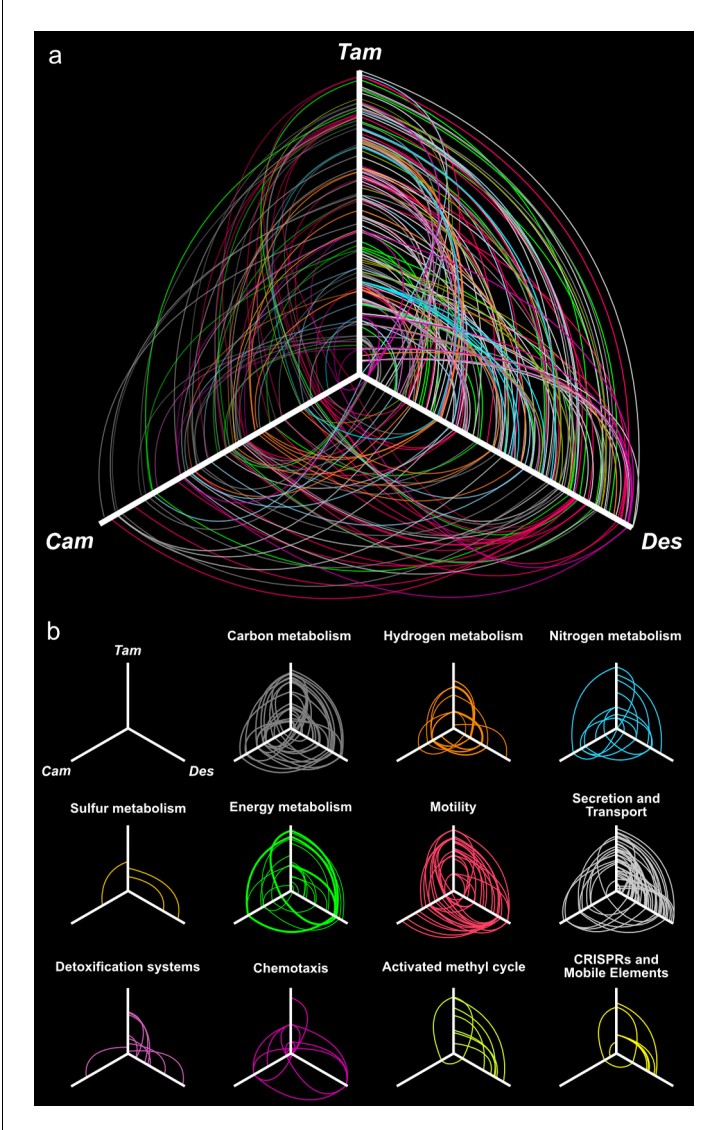

**Figure 7.** Hives plot presenting the comparative genomic analysis of the *T. ammonificans* (Tam) genome with the closest relative available genomes of *D. thermolithotrophum* (Des) and the ecologically similar *Epsilonproteobacterium C. mediatlanticus* (Cam). The three axes represent the organism's linearized genomes with the origin of replication at the center of the figure. Genome size was normalized for visualization purpose. (**A**) Localization of homologous genes and syntenic regions among the three genomes. Lines colored according to the general pathway code adopted in *Figure 3* connect homologous coding sequences. Lines opacity is proportional to similarity between sequences (darker = higher similarity), line width is proportional to the extent of the syntenic area. (**B**) Hives panel of the homologous genes divided by pathways. Collinear lastZ alignments and interactive dot-plot replicating the analysis are accessible at the GenomeEvolution website with the following permanent addresses: *T. ammonificans* and *D. thermolithotrophum* (http://genomevolution.org/r/9vb3); *T. ammonificans* and *C. mediatlanticus* (http://genomevolution.org/r/9vb4); *D. thermolithotrophum* and *C. mediatlanticus* (http://genomevolution.org/r/9vb8).

and the alpha, beta, gamma and delta subunits of the Group 3b multimeric cytoplasmic Hyd sulfhydrogenase (*hydABCD*, Theam_0484–0487). The second hydrogenase cluster in the genome codes for the maturation protein HypF (*hypF*, Theam_1116), the three subunit of the Group 3a cytoplasmic F420-reducing hydrogenase Frh (*frhACB*, Theam_1117–1119) and the Group 1 hydrogenases Hyn (*hynABC*, Theam_1121–1123) with associated maturation factors/chaperones Hyp (Theam_1124–

1128). We also found other hydrogenase maturation factors (Theam_0922–0924) and the formate dehydrogenase Fdh (*fdhABC*, Theam_1020 and Theam_0966–0967 respectively) scattered in the genome.

We reconstructed the phylogeny of *T. ammonificans* [NiFe]-hydrogenases (*Figure 5*), and we assigned them to known classes of hydrogenases (Groups 1 through 4) according to their phylogenetic position in the tree and the presence of known conserved amino acid motifs typical of each group (*Vignais and Billoud, 2007*). Group 1 [NiFe]-hydrogenases are normally coupled to anaerobic respiration through electron transfer to the membrane quinone and menaquinone pool. Comparative analyses of the hydrogenases revealed that the Group 1 cluster in *T. ammonificans* (Hyn) is more similar to the hydrogenases found in *Deltaproteobacteria* than to homologs in other *Aquificae* (*Figure 5*). This finding raises interesting questions about the origin of the hydrogen oxidation Group 1 enzymes in *T. ammonificans* and suggests that an event of lateral gene transfer occurred between *T. ammonificans* and the *Deltaproteobacteria*. Further, the periplasmic nitrate reductase (*napA*) and the *nap* operon structure in *T. ammonificans* also appears to be closely related to that of *Deltaproteobacteria*, suggesting that a horizontal gene transfer event involved the entire hydrogen oxidation (Hyn)/nitrate reduction respiratory pathway (see discussion).

Contrarily to members of the *Aquificaceae* and *Hydrogenothermaceae*, *T. ammonificans* and *D. thermolithotrophum* genomes code for both Group 3 cytoplasmic hydrogenases, Group 3a F420-reducing hydrogenases (*frhACB*) and Group 3b multimeric cytoplasmic sulfhydrogenases (*hydABCD*; *Figure 5*; *Jeon et al., 2015*). Comparative analyses of these two clusters revealed that other members of the *Desulfurobacteriaceae* family were the closest relatives, while the next closest enzymes were those found in hyperthermophilic *Euryarchaeota* (*i.e. Pyrococcus abyssi*) and in methanogens (*Figure 5*). Overall, these results imply a polyphyletic origin for the group 3 cytoplasmic hydrogenases. Comparative analyses revealed similarities with hyperthermophilic *Euryarchaeota* and methanogens (*Figure 5*), suggesting a possible common ancestry for the group 3 cytoplasmic hydrogenases of *Archaea* and *Desulfurobacteriaceae*. Together with Group 3b sulfhydrogenases and Group 4 Ech, F420-reducing hydrogenases are in fact typically found in methanogens in which the F420-cofactor is used as an electron carrier in the reduction of $CO_2$ or other C1 compounds to methane (*Vignais and Billoud, 2007*). Cultures of *T. ammonificans* grown with nitrate as the terminal electron acceptor did not present the typical autofluorescence of methanogens due to F420-cofactor fluorescence when observed under UV in epifluorescence microscopy and the genome lacks the gene involved in the biosynthetic pathway of the F420-coenzyme. Due to these observations, we suggest that these hydrogenases may work in conjunction with ferredoxin in *T. ammonificans*. Group 3b Hyd sulfhydrogenases appear to be of mixed origins despite being present in a single operon. Three of the subunits share similarities with the *Thermococcales* (*Euryarchaeota; hydBCD*) while the catalytic subunit (HydA) appears to be unique to the *Desulfurobacteriacea*, not having any other significant homolog in the database (*Figure 6*). Comparative analyses indicated that Ech shared similarity with *Epsilonproteobacteria* Group 4 hydrogenases (*Figure 5*). Interestingly, Group 2 hydrogenases are found in microaerobic members of the *Aquificaceae* and *Hydrogenothermaceae* and in microaerobic *Epsiloproteobacteria* but are not found in anaerobes, suggesting a possible link with oxygen adaptation. In contrast, Group 3a is found in strict anaerobes, including *T. ammonificans* and methanogens (*Figure 5*).

Finally, Ech hydrogenases are the only Group 4 hydrogenases found in *T. ammonificans*. Both the reductive TCA cycle and the reductive acetyl-CoA pathway involve reaction steps that are driven by reduced ferredoxin (*Fuchs, 2011*). In the case of *T. ammonificans,* the enzymes formate dehydrogenase, CO-dehydrogenase, pyruvate synthase and 2-oxoglutarate synthase require reduced ferredoxin. We hypothesize that *T. ammonificans* can accomplish ferredoxin reduction in two different ways: the first involves cytosolic hydrogenases (Group 3, Frh and Hyd; *Figure 3*) commonly found in methanogens, where they reduce ferredoxin via electron bifurcation (*Fuchs, 2011*); the second involves membrane bound hydrogenases (Group 4, Ech; *Figure 3*). With the exception of the F420-reducing (Frh) hydrogenases, representatives of the remaining three groups were identified in the proteome of *T. ammonificans* (*Figure 3*).

## Nitrate ammonification

*T. ammonificans* HB-1 can use nitrate as a terminal electron acceptor and reduces it to ammonium (*Vetriani et al., 2004*). Periplasmic nitrate reductases (Nap) have been studied extensively in some

*Proteobacteria*, and are comprised of the catalytic enzymes – typically a heterodimer (NapAB) – associated with various other subunits involved in channeling electrons to the NapA reactive center (*Moreno-Vivián et al., 1999*; *Vetriani et al., 2014*). In *T. ammonificans*, the genes encoding for the Nap complex are organized in a single operon, *napCMADGH*. Comparative genomic analyses with genomes from *Aquificae* and *Proteobacteria* indicate that the *nap* operon structure of *T. ammonificans* is unique to this organism, and shares similarity with *Desulfovibrio desulfuricans* and other *Deltaproteobacteria*. All other *Aquificae* whose genomes are available have a Nap operon similar to that of *Epsilon-* and *Gammaproteobacteria* (*Vetriani et al., 2014*). Moreover, unlike other members of the *Aquificae*, the nitrate reductase of *T. ammonificans* (encoded by gene Theam_0423, expressed in the proteome; *Figure 3*) appears to be of the monomeric type (*sensu*; *Jepson et al., 2006*), and the NapB subunit is missing (*Figure 3*). We also identified a putative nitrite reductase-encoding gene possibly involved in the reduction of nitrite to ammonium (NirA; Theam_1000, *Figure 3*), which was also expressed in the proteome (*Figure 3*). However, the exact mechanism through which *T. ammonificans* reduces nitrite to ammonium remains to be experimentally elucidated.

## Sulfur reduction

While *T. ammonificans* is able to reduce elemental sulfur to hydrogen sulfide (*Vetriani et al., 2004*), the sulfur reduction pathway remains unclear. We identified in the genome a putative polysulfide reductase of the sulfide-quinone reductase family (*sqr*, Theam_0841, *Figure 3*). We propose that this membrane-bound complex can utilize polysulfide formed from the reaction of $S^0$ with sulfide (naturally enriched in hydrothermal systems) via quinone oxidation (*Figure 3*). The putative *sqr* gene of *T. ammonificans* shares similarities with other *Aquificae* and *Epsilonproteobacteria* (average 56% similarity to both groups). In particular, the *sqr* gene appears to be conserved also in *Caminibacter mediatlanticus* (*Figure 7*; *Giovannelli et al., 2011*), whose sulfur reduction pathway is yet to be elucidated. Another possibility is that *T. ammonificans* uses a NAD or FAD-dependent reductase (*nsr*, Theam_1321) to reduce sulfur, a mechanism previously described in the sulfur-reducing archaeon *Pyrococcus furiosus* (*Schut et al., 2013*, *2007*). However, the putative polysulfide reductase of *T. ammonificans* has only about 33% identity to homologous enzymes of sulfur reducing bacteria and archaea identified in the database.

## Resistance to oxidative stress

*T. ammonificans* is a strict anaerobe. However, its genome codes for genes involved in the detoxification of oxygen radicals, including a catalase/peroxidase (Theam_0186), whose activity was previously detected (*Vetriani et al., 2004*), a putative superoxide reductase (Theam_0447), a cytochrome-C peroxidase (Theam_1156) and a cytochrome bd complex (Theam_0494–0496, *Figure 3*). The latter three enzymes were identified in the proteome (*Figure 3*). The cythochrome bd complex has been shown to contribute to oxygen tolerance in other anaerobic bacteria (*Das et al., 2005*), and to contribute to the detoxification of nitrous oxide radicals (*Mason et al., 2009*).

## Energy conversion

The genome of *T. ammonificans* contains two gene clusters coding for the NADH dehydrogenase (Theam_0731–0745 and Theam_1488–1498). The genes appear to have undergone an inversion during the duplication process and the second copy is missing four genes in the middle of the operon (*nuoEFG* and a transcriptional regulator). The two clusters share on average 60% gene similarity. Only one copy of the gene for the synthesis of ATP synthetase complex was present (Theam_1605–1606 and Theam_1656–1662). Three different hydrogen-sodium symporters (*nhe*, Theam_0443, 0413 and 0647) were present with low pairwise similarity. NADH dehydrogenase second cluster and ATP synthetase complex appear to be conserved in the close relative *D. thermolithotrophum* and in the *Epsilonproteobacterium*, *C. mediatlanticus* TB-2 (*Figure 3*).

## Motility, cell sensing and biofilm formation

*T. ammonificans* posses one to two terminal flagella (*Vetriani et al., 2004*). The genes involved in flagella formation and assembly are prevalently organized in a single large cluster in the genome (from Theam_1440 to Theam_1468, *Figure 7*), putatively organized in three distinct operons. The

hook-filament junction proteins genes *flgK* and *flgL* (Theam_1384 and Theam_1385), the filament and filament cap proteins *fliC* and *fliD* (Theam_1160 and Theam_1162) together with few flagellar maturation factors and motor switch genes (*motA* and *motB*, Theam_1087 and Theam_1085–1086, respectively) are instead scattered in the genome. The entire cluster of genes shares similarities with *D. thermolitotrophum,* although the similarity for some of the proteins is as low as 50%. Despite this, comparative analyses revealed a similar organization in *D. thermolitotrophum* where the cluster appears inverted in the same relative genomic position and constituted one of the main region of synteny (*Figure 7b*). By contrast, the flagellar genes in the *Epsilonproteobacterium, Caminibacter mediatlanticus* TB-2, are scattered throughout the genome (*Figure 7b*). Numerous genes for adhesion and pili were present, generally organized in small clusters (*Figure 7b*).

Four different putative methyl-accepting chemotaxis sensory transducer proteins were found in the genome (*mcp*, Theam_0157, 0165, 0845 and 1027). One of those, Theam_0845, is surrounded by receptors *cheW* and *cheA*, and by the response regulator *cheY*. The entire group is organized in a single operon, *cheYVBWAZ* conserved in *D. thermolithotrophum* and other *Aquificales* with the exception of *cheW*, which shares higher similarities with the *Epsilonproteobacterium, C. mediatlanticus* TB-2 (*Figure 6*). We failed to find in the genome of *T. ammonificans* a homolog of the primary receptor encoding gene *cheR* (*Wadhams and Armitage, 2004*). We identified in the genome numerous membrane transporters for molybdenum (*modCBA*, Theam_0787–0789), iron (*fhuDBC*, Theam_0192–0194), zinc (*znuAB*, Theam_0238 and 0689) and cobalt/nickel (*cbiOQMN*, Theam_1522, 1523, 1525 and 1526) all involved in the uptake of important micro-elements for enzyme and cofactor synthesis. We also identified the complete type II secretion pathway, which is conserved in gram-negative bacteria and responsible for protein and toxin translocation to the extracellular milieu. Proteins secreted by the type II pathway include proteases, cellulases, pectinases, phospholipases, lipases, toxins and in some cases type III and IV pili (*Douzi et al., 2012*). In some bacteria, the expression of secretion type II pathway genes are regulated either by quorum-sensing or by the environmental factors at the site of colonization (*Douzi et al., 2012*; *Sandkvist, 2001*). Biofilm formation in bacteria is often linked to quorum-sensing mechanisms, regulating the settlement of planktonic cells in relationship to environmental sensing, substrate suitability and cell densities. It is likely that *T. ammonificans* maintains a mostly attached lifestyle in hydrothermal vent environments due to the high turbulence associated with fluid flux and mixing with seawater. We speculate that the type II pathway genes, together with chemotaxis and flagellar/adhesion genes may play a key role in the formation of biofilm. Extracellular secreted proteins may in fact be used to 'condition' the colonized substrate and to interact/remodel existing and newly formed biofilms.

## Activated methyl cycle

The activated methyl cycle is a conserved pathway in which S-methyl groups from L-methionine are converted into a biochemically reactive form through insertion into an adenosyl group (*Vendeville et al., 2005*). S-adenosyl-L-methionine provides activated methyl groups for use in the methylation of proteins, RNA, DNA and certain metabolites central to the core cell machinery. The activated methyl cycle exists in two recognized forms, involving respectively *luxS* or *sahH* genes (*Vendeville et al., 2005*). Genome-based reconstruction of the activated methyl cycle indicated that, in *T. ammonificans,* involves the *sahH* gene, while a *luxS* homolog is absent (*Figure 3—figure supplement 2*). Methionine methyl groups are adenosylated by a methionine adenosyltransferase (*samS*, Theam_1075). The methyl group is thus activated and readily available for the S-adenosylmethionine-dependent methyltransferase (*samT*) for methylation of substrate. In *T. ammonificans*, *samT* is a pseudogene, as it contains two stop codons (Theam_0630). Homologs of *samT* are absent in the genome of the other *Aquificae* and the pathway appears to be incomplete. Only in *P. marina* and *D. themolithotrophum* the functionally equivalent DNA-cytosine methyltransferase (*dcm*) is present. The presence of a gap in such a central pathway in the *Aquificae* and the retrieval of a *samT*-like pseudogene in *T. ammonificans* raises interesting question on the functioning of the activated methyl cycle in those organisms. An interesting working hypothesis is that enzyme thermal stability at the physiological temperature of these bacteria might have driven the loss of *samT* and the replacement by a not yet identified alternative enzyme in most *Aquificae*.

**Table 3.** List of the genomes belonging to the *Aquificae* phylum used for comparative genomic analyses.

| Organism | Genome acc. number | Genome length | GC content | Num. genes |
|---|---|---|---|---|
| *Aquifex aeolicus* VF5 | NC_000918 | 1,590,791 | 43% | 1782 |
| *Desulfurobacterium* sp. TC5-1 | NZ_ATXC01000001 | 1,653,625 | 40% | 1680 |
| *Desulfurobacterium thermolithotrophum* DSM 11699 | NC_015185 | 1,541,968 | 35% | 1561 |
| *Hydrogenivirga* sp. 128–5 R1-1 | NZ_ABHJ01000551 | 3,038,240 | 44% | 3756 |
| *Hydrogenobacter thermophilus* TK-6 | NC_013799 | 1,743,135 | 44% | 1897 |
| *Hydrogenobaculum* sp. Y04AAS1 | NC_011126 | 1,559,514 | 35% | 1631 |
| *Persephonella marina* EX-H1 | NC_012440 | 1,983,966 | 37% | 2067 |
| *Persephonella* sp. IF05-L8 | NZ_JNLJ01000001 | 1,828,858 | 35% | 1920 |
| *Sulfurihydrogenibium azorense* Az-Fu1 | NC_012438 | 1,640,877 | 33% | 1722 |
| *Sulfurihydrogenibium* sp. YO3AOP1 | NC_010730 | 1,838,442 | 32% | 1832 |
| *Sulfurihydrogenibium subterraneum* DSM 15120 | NZ_JHUV01000001 | 1,610,181 | 32% | 1701 |
| *Sulfurihydrogenibium yellowstonense* SS-5 | NZ_ABZS01000228 | 1,534,471 | 33% | 1637 |
| *Thermocrinis albus* DSM 14484 | NC_013894 | 1,500,577 | 47% | 1631 |
| *Thermocrinis ruber* DSM 23557 | NZ_CP007028 | 1,521,037 | 45% | 1625 |
| *Thermocrinis* sp. GBS | NZ_JNIE01000001 | 1,315,625 | 41% | 1417 |
| *Thermosulfidibacter takaii* ABI70S6 | NZ_AP013035 | 1,816,670 | 43% | 1844 |
| *Thermovibrio ammonificans* HB-1 | NC_014926 | 1,759,526 | 52% | 1820 |

## Discussion

### Evidence for a reductive acetyl-CoA pathway in t*he Desulfurobacteraceae* lineage

The reductive acetyl-CoA pathway is present in acetogenic bacteria, methanogens, sulfate-reducers and anammox bacteria (*Berg, 2011*; *Fuchs, 2011*; *Hügler and Sievert, 2011*). It is believed to be the oldest carbon fixation pathway on Earth, due to the centrality of acetyl-CoA in metabolism, the low energy requirement (~1 ATP required *vs* 2–3 ATP for the rTCA cycle) and the limited necessity of de novo protein assembly (*Berg et al., 2010*; *Fuchs, 2011*). We hypothesize that the reductive acetyl-CoA pathway is used as an additional or alternative pathway to fix $CO_2$ in *T. ammonificans*.

The finding of a potentially active reductive acetyl-CoA pathway (reactions 1 to 6; *Figure 3*) in *T. ammonificans* and other members of the *Desulfurobacteraceae* is intriguing. The pathway is missing reaction 2, the formyl-THF synthesis for which no obvious 10-formyl-THF synthetase was found (*Figure 3*). However, the 10-formyl-THF synthetase of the reductive acetyl-CoA pathway may have been replaced by alternative enzymes, such as 5-formyl-THF cycloligase (Theam_1206) or 10-formyl-THF deformylase (Theam_0826) working in reverse (*Table 2*, *Figure 3*). Promiscuous enzymes that function in more than one pathway have been long hypothesized and recently described in *Escherichia coli* (*Jensen, 1976*; *Kim et al., 2010*). Assuming low substrate specificity, other candidate enzymes with similar substrate requirements are the phosphoribosylglycinamide formyltransferase (Theam_1211) or phosphoribosylaminoimidazolecarboxamide formyltransferase/IMP cyclohydrolase (Theam_0328), which could catalyze the synthesis of formyl-THF, albeit with sub-optimal kinetics. While all four enzymes were detected in the proteomic analysis (*Figure 3*), we think that the possible involvement of a 5-formyl-THF cycloligase in the synthesis of formyl-THF in *T. ammonificans* is particularly appealing, as this would provide an evolutionary link between the $N^5$-formate uptake of methanogens and $N^{10}$-formate uptake in acetogens (*Braakman and Smith, 2012*). Despite the absence of the 10-formyl-THF synthetase, the genome of *T. ammonificans* encodes and expresses a type V CodH (*Figure 3*).

We investigated the available genomes of other members of the *Aquificae* for the presence of genes homologous to the putative reductive acetyl-CoA pathway of *T. ammonificans* (*Table 2*). The similarity of the reductive acetyl-CoA pathway-related genes of *T. ammonificans* to those from organisms outside the *Desulfurobacteraceae* is low. The type V *codH* gene identified in *T.*

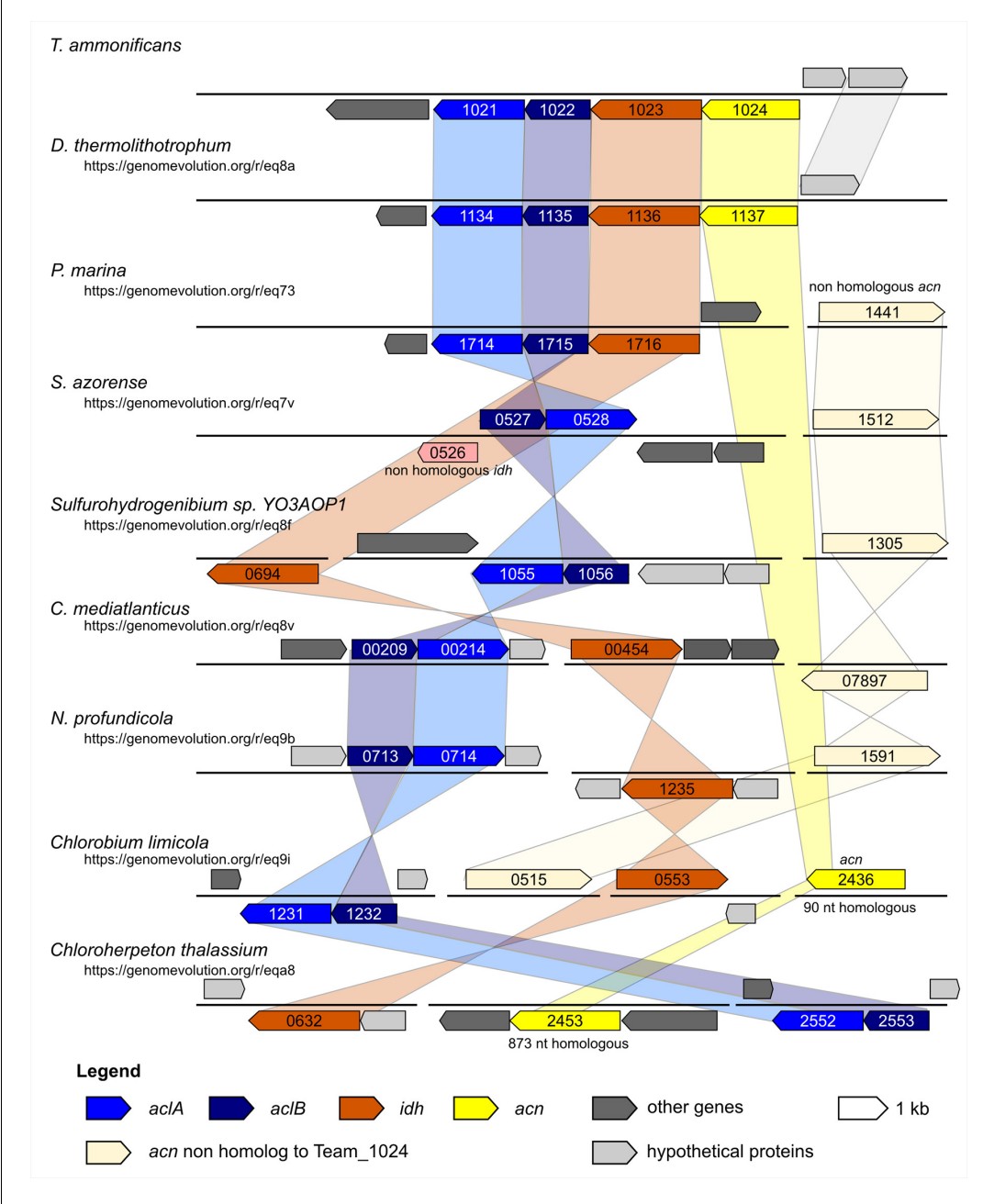

**Figure 8.** Syntheny diagram presenting the genome organization around the rTCA key enzyme ATP citrate lyase. In *T. ammonificans* the two subunits of the ATP citrate lyase enzyme are organized in a single operon together with the isocitrate dehydrogenase and the aconitate dehydratase. The numbers inside each gene represent the locus number for the organism. Shaded color connects synthenic regions. The website address below each organism name is a permanent link to the pairwise analysis performed on the Genome Evolution server (http://genomevolution.org/) *against* T. ammonificans.

ammonificans is found exclusively in members of the family *Desulfurobacteriaceae*, with exception of a low similarity homolog present in *Persephonella marina* (41% amino acid identity) (*Table 2*). Furthermore, most of the genes involved in reaction 3, 4 and 5 (*Figure 3*) have homologs in other *Aquificae* or are substituted by functional equivalents with low amino acid identity (<40%). To understand the relationship of the *T. ammonificans* CodH with the homologous enzymes of methanogens, homoacetogens and sulfate-reducers, we reconstructed the phylogenetic history of these enzymes

with a particular focus on type V CodH (*Figure 4*). Sequences from the *Desulfurobacteriaceae* appear to be related to those present in the *Archaea, Ferroglobus placidus* and *Archaeoglobus* spp., and similar to *Methanosarcina* spp. and homoacetogenic bacteria. The CodH found in CO-utilizing bacteria is only distantly related to the one found in *T. ammonificans* (*Techtmann et al., 2012*), while the CodH sequence of *Persephonella marina* appears to be one of a kind, with the enzyme of *Ferroglobus placidus* as its closest relative (44% similarity). Taken together, these findings suggest that, within the *Aquificae*, the type V CodH is exclusive to the *Desulfurobacteriaceae*.

The retrieval of the CodH enzyme from the proteome of *T. ammonificans* (*Figure 3*, *Figure 3— source data 1*), along with the catalytic subunit of the formate dehydrogenase (FdhA), suggests that the reductive acetyl-CoA pathway could be operational in *T. ammonificans* in addition to the rTCA cycle. The simultaneous presence of two distinct carbon fixation pathways has been confirmed so far only in the endosymbiont of the deep-sea vent tubeworm *Riftia pachyptila* (*Markert et al., 2007*). Recent findings suggest that the *Riftia* symbiont can switch from the Calvin-Benson-Bassham (CBB) to the rTCA cycles in response to low-energy supply (*e.g.,* low concentrations of hydrogen sulfide in the vent fluids, being the rTCA cycle more energetically favorable than the CBB cycle; *Markert et al., 2007*). Genomic evidence of the simultaneous presence of different carbon fixation pathways came also from genome analysis of *Ammonifex degensii* and *Ferroglobus placidus* (*Berg, 2011*; *Hügler and Sievert, 2011*), where the reductive acetyl-CoA pathway *seems to be coupled to an incomplete CBB cycle. The use of different carbon fixation strategies could be advantageous under energy limiting conditions (e.g.,* deep biosphere), could optimize overall carbon fixation or provide different precursors for biosynthetic pathways, and could be more widespread than originally thought.

Further laboratory analyses will identify the exact role of the type V CodH in *T. ammonificans* and the potential contribution of the reductive acetyl-CoA to overall carbon fixation.

## Comparative genomic of *T. ammonificans*: ancestral and acquired metabolic traits

The genome of *T. ammonificans* HB-1 displays a large degree of mosaicism (*Figure 6*). When comparing the coding sequences (CDS) of *T. ammonificans* with available genomes, 34% of the CDS shared higher similarity to genes outside the *Aquificae,* suggesting that these genes were acquired from more distantly related taxa.

We carried out comparative genomic analyses among *T. ammonificans* and all available *Aquificae* genomes (*Table 3*). *Desulfurobacterium thermolithotrophum* DSM 11699 (its closest relative whose genome was sequenced) and *Caminibacter mediatlanticus* TB-2 (an *Epsilonproteobacterium* which, albeit phylogenetically distant, shares the same physiology and occupies a similar ecological niche as *T. ammonificans,* only at lower temperature; *Figure 7*) were further used as a direct comparison. We identified areas of synteny between *T. ammonificans* and the closely related *D. thermolithotrophum* in the region surrounding the genes encoding for key enzymes responsible for the citrate cleavage in the rTCA cycle (*Figure 7b* and *Figure 8*). This region is conserved also within the *Hydrogenothermaceae* (in particular *P. marina*) and members of the *Epsilonproteobacteria*.

The conserved regions between the genomes of *T. ammonificans* and *D. thermolithotrophum* included the two hydrogenase clusters, the flagellum and the NADP dehydrogenase complex. The order and position of the hydrogenase and NADP dehydrogenase clusters were highly conserved also in *C. mediatlanticus*, while flagellar genes were scattered throughout the genome (*Figure 7*). These results suggest that the genes encoding hydrogen utilization and functions related to energy conversion are among the oldest genomic regions present in those organisms, and overall are conserved across phyla. When we analyzed the central metabolism of *T. ammonificans* using comparative genomic approaches, the presence of two distinct groups of genes became evident: the first group was related to early-branching bacterial or archaeal lineages and coded for enzymes involved in several central metabolic pathways, while the second group of genes appeared to have been acquired later. Among the first group of genes, we identified: (I) the cytoplasmic [Ni-Fe]-hydrogenases Group 3 that were related to the enzymes found in methanogens and thermophilic *Euryarchaeota,* in which they are involved in ferredoxin reduction (*Figure 3* and *Figure 8*); (II) sulfur-reduction related genes (*Figure 3*); and (III) the genes coding for the enzymes of the reductive acetyl-CoA pathway (*Figure 3*). These pathways are either present in early branching *Archaea*, or are directly involved in metabolic reactions that do not require oxygen (or oxygen by-products) and use substrates of geothermal origin that are likely to have been abundant on Early Earth. Moreover, most

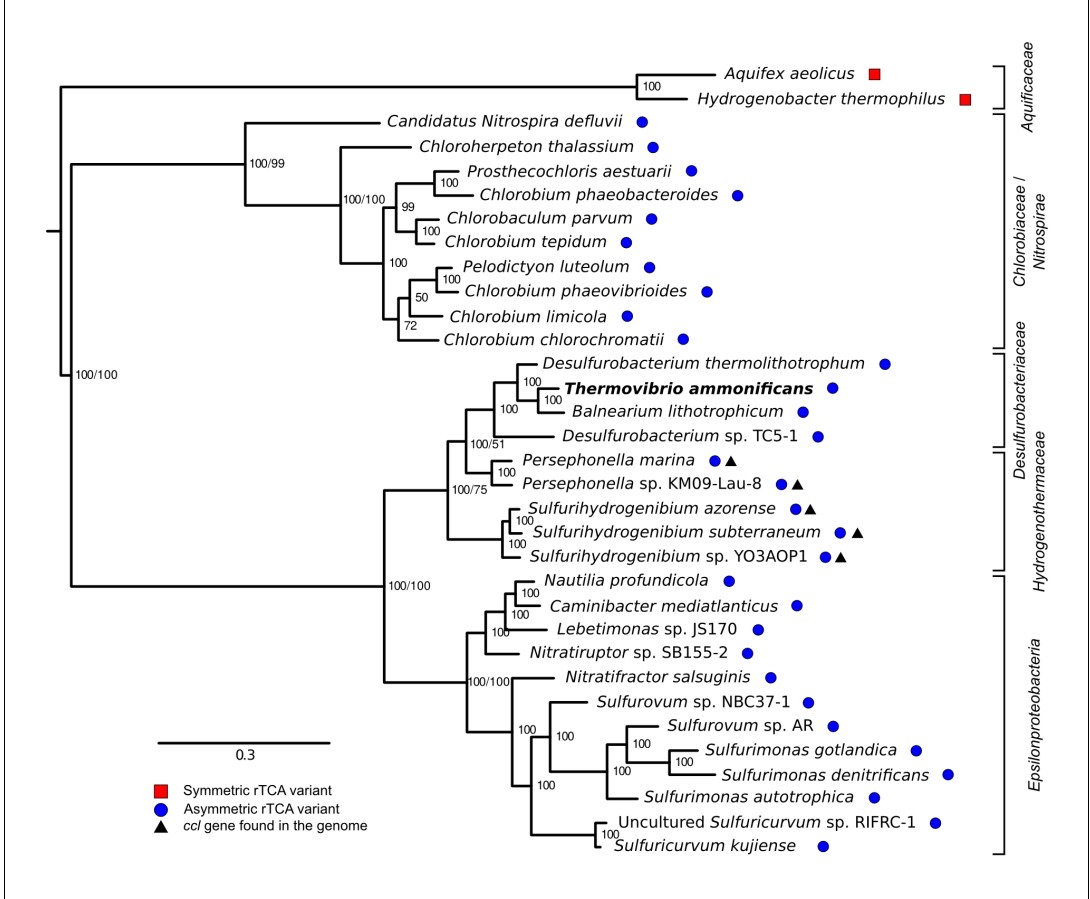

**Figure 9.** Phylogenetic tree of ATP citrate lyase amino acid sequences. The tree was constructed with Bayesian inference and maximum likelihood methods and presents the phylogenetic relationship among the concatenated subunit of the ATP citrate lyase in different organisms known to use the rTCA cycle. The numbers near the nodes represent the bayesian posterior probability (left number) and the maximum-likelihood confidence values based on 1000 bootstrap replications (right number). Bar, 30% estimated substitutions. Accession numbers are reported in *Figure 9—source data 1*. Squares – symmetric rTCA cycle variant characterized by the presence of two-step enzyme reactions for the cleavage of citrate and the carboxylation of 2-oxoglutarate. Circles – asymmetric rTCA cycle variant characterized by one-step reaction enzymes catalyzing the above reactions. Triangles – presence of a citryl-CoA lyase in the genome. See Supplementary Materials for details.

The following source data is available for figure 9:

**Source data 1.** Accession number of the ATP citrate lyase sequences used for the reconstruction of the phylogenetic history of the enzyme presented in *Figure 2* in the main text.

key reactions in these pathways are catalyzed by enzymes that are extremely sensitive to oxygen. Our findings are consistent with a recent reconstruction of the ancestral genome of the last universal common ancestor (LUCA), which suggest that LUCA was a hydrogen-dependent autotroph capable of S utilization that could fix $CO_2$ via the reductive acetyl-CoA pathway (*Weiss et al., 2016*). Taken together, these observations suggest that part of the central metabolism of *T. ammonificans* is ancestral and emerged prior to oxygenic photosynthesis.

By contrast, some genes and associated metabolic pathways appear to have been acquired by the *T. ammonificans* lineage at a later time. One example is the ability to conserve energy by nitrate reduction. The hypothesis that nitrate respiration is an acquired trait in *T. ammonificans* is supported by the structure of the *nap* operon. The monomeric *napA* gene of *T. ammonificans* is homologous to that of *Desulfovibrio desulfuricans* and other *Deltaproteobacteria* (*Figure 6*; online interactive version) and apparently has been acquired laterally. Furthermore, it shares high similarity with assimilatory nitrate reductases (Nas) and it could be an evolutionary intermediate form between assimilatory

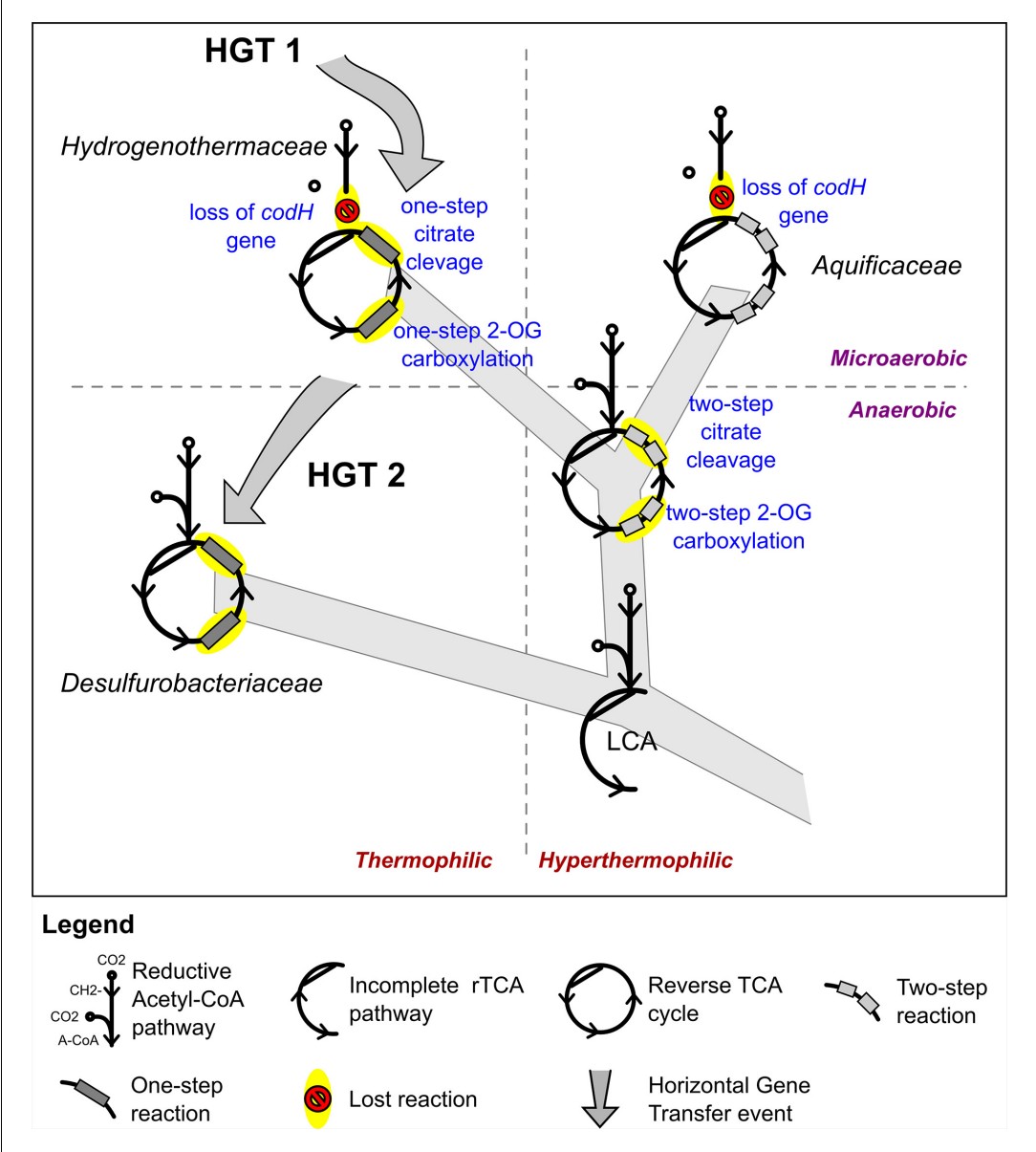

**Figure 10.** Proposed evolution of the carbon fixation pathway in the *Aquificae* phylum. Proposed evolution of the carbon fixation pathway in the *Aquificae* phylum and reconstruction of the ancestral carbon fixation for the last common ancestor (LCA) based on the results of integrated phylogenetic, comparative genomic and phylometabolic analyses.

Nas and the dimeric respiratory Nap. In line with this observation, phylogenetic analyses suggest that the membrane bound [Ni-Fe]-hydrogenases of Group 1 and 4 are of delta/epsilonproteobacterial origin (*Figure 5*). Based on the finding that Group 1 Hyn hydrogenases were expressed when *T. ammonificans* was grown under nitrate-reducing conditions (*Figure 3*), we suggest that these enzymes are linked through the membrane quinone pool to nitrate reduction and may have been acquired simultaneously with the *nap* genes.

A second example is represented by the oxygen radical detoxification enzymes encoded by the genome of *T. ammonificans*. We propose that such genes are not part of the core, or ancestral genome of *T. ammonificans*, but that they evolved as a response to exposure to toxic oxygen radicals. In the modern ocean, catalase and peroxidase may provide protection to oxygen-sensitive enzymes during transient exposure to oxygen associated with entrainment of deep seawater in

hydrothermal fluids, or when the organism is displaced into the water column. This second scenario may have important implication for the dispersal of vent microorganisms to other vent sites. We hypothesize that *T. ammonificans* is able to deal with oxygen exposures using two different mechanisms: the first response mechanism would be activated following exposure to oxygen at physiological temperatures (60–80°C). In that case, the oxygen-stress related genes are induced, protecting the cell machinery against damages. The second mechanism would be a response to prolonged exposure to oxygen at temperatures below the physiological threshold. Such prolonged exposure would trigger a metabolic shut-down, enabling survival of the organism in cold seawater. The latter hypothesis is supported by experiments where batch cultures of *T. ammonificans* were exposed for a prolonged time at 4°C, which revealed the survival of the bacterium despite being exposed to oxygen (data not shown).

In conclusion, the nitrate reduction pathway, as well as oxygen stress-related genes, may represent adaptations acquired by the *T. ammonificans* lineage to cope with the rise of oxygen on Earth.

## Insight into the evolution of carbon fixation

Six different pathways of carbon fixation are known to date (*Fuchs, 2011*). In the modern biosphere, the Calvin-Benson-Bassham cycle is the dominant mechanism of $CO_2$ fixation. Yet, other anaerobic pathways have been investigated and are proposed to represent the ancestral autotrophic carbon fixation pathway. In particular, the reductive acetyl-CoA pathway is thought to be among the earliest carbon fixation pathways that have emerged on Earth (*Fuchs, 2011*). This hypothesis is supported by numerous observations: (I) the presence of this pathway in the early branching methanogenic archaea; (II) its low energy requirements and low need of de novo protein synthesis; (III) its capacity to incorporate different one-carbon compounds and carbon monoxide of geothermal origin, and (IV) the extreme oxygen sensitivity of its key enzymes which have common roots in *Bacteria* and *Archaea* (reviewed in *Berg et al., 2010*; *Fuchs, 2011*; *Hügler and Sievert, 2011*). Despite differences in their structure, all extant carbon fixation pathways can be theoretically derived from a putative rTCA cycle/reductive acetyl-CoA hybrid pathway, as proposed by a recent phylometabolic reconstruction of the evolution of carbon fixation (*Braakman and Smith, 2012*). *According to this study, both Aquificae and acetogenic bacteria represent the closest living example of the archetypal network, and both diverged from the pre-LUCA pathway under the selective pressure of energy-efficiency (acetogens) and oxygen sensitivity (Aquificae).* Further, a recent reconstruction of the genome of LUCA based on gene phyletic pattern reconstruction is consistent with some of these findings and suggests that LUCA's genomic makeup point to autotrophic acetogenic and methanogenic roots and to the ancestry of the reductive acetyl-CoA pathway (*Weiss et al., 2016*).

The presence of a CodH type V enzyme in the genome of *T. ammonificans* and other members of the *Desulfurobacteriaceae*, and the presence of a type II CodH in *Desulfurobacterium thermolithotrophum* suggest that the reductive acetyl-CoA pathway could be operational in extant members of the *Desulfurobacteriaceae*. Furthermore, this finding supports the scenario proposed by Braakman and Smith that a complete and operational reductive acetyl-CoA pathway was present in the ancestor of the phylum *Aquificae* (*Braakman and Smith, 2012*). Comparative analyses revealed that the CodH enzyme is conserved only within the *Desulfurobacteriaceae*, consistent with the strict anaerobic nature of the members of this family and the extreme oxygen sensitivity of CodH. The rise of oxygen has been interpreted as one of the factors responsible for the diversification of carbon fixation from the ancestral pathway, and could explain the subsequent loss of the CodH in the generally facultative microaerobic *Hydrogenothermaceae* and *Aquificaceae* within the *Aquificae* (*Table 2*).

Comparative genomic analyses of rTCA cycle genes allow the description of a possible evolutionary scenario for the rTCA cycle in *Aquificae*. Members of the *Aquificaceae* (e.g., *Aquifex aeolicus*; *Figure 9* and *Figure 3—figure supplement 1*) have the two-step version of the rTCA cycle (*Figure 3—figure supplement 1A*), where citrate cleavage is accomplished by the combined action of the enzymes citryl-CoA synthetase and citryl-CoA lyase (encoded by the *ccl* gene), and the carboxylation of 2-oxoglutarate is catalyzed by the two enzymes 2-oxoglutarate carboxylase and oxalosuccinate reductase (*Aoshima et al., 2004*; *Braakman and Smith, 2012*). Since citryl-CoA synthetase and 2-oxoglutarate carboxylase likely evolved by duplication of the genes for the succinyl-CoA synthetase and pyruvate carboxylase (*Aoshima, 2007*; *Braakman and Smith, 2012*), a complete rTCA cycle could have evolved in the *Aquificaceae* from an ancestral, incomplete version of the cycle. In contrast, the two other groups of *Aquificae*, the *Hydrogenothermaceae* and the

*Desulfurobacteriaceae* (*Figure 1—figure supplement 1*), use the one-step – and more recent – version of the rTCA cycle (*Figure 10*, *Figure 9* and *Figure 3—figure supplement 1B*), involving ATP citrate lyase (ACL, encoded by the *acl* gene) and isocitrate dehydrogenase, that carry out citrate cleavage and 2-oxoglutarate carboxylation in two single enzyme reactions, respectively (*Braakman and Smith, 2012*).

The enzyme responsible for citrate cleavage, the ATP citrate lyase, likely evolved through gene fusion of the genes of CCS and CCL (*Aoshima et al., 2004*). Phylogenetic analyses of ACL suggests that this gene fusion event did not happen within the *Aquificae,* as *Nitrospira* and *Chlorobia* have an evolutionary older version of ACL than *Hydrogenothermaceae* and *Desulfurobacteriaceae* (*Figure 9*; *Hügler et al., 2007*). Hence, it is likely that these two groups acquired ACL through HGT (*Figure 9* and *Figure 8*; *Hügler et al., 2007*). Furthermore, our comparative analysis showed that: (I) in the *Hydrogenothermaceae* and in the *Desulfurobacteriaceae*, the enzymes of the first half of the rTCA cycle, as well as aconitase (*acnA*), share a common ancestor with the *Aquificaceae* and can be considered part of the core genome of the phylum *Aquificae*, while the remaining enzymes are either the result of gene duplication or have been acquired by horizontal gene transfer (*Hügler et al., 2007*); (II) the two-step citrate cleavage is exclusive to the *Aquificacea* (*Figure 9*); (III) the *ccl* gene is still present in the *Hydrogenothermacea* (in addition to *acl*), and one sequenced strain of the *Hydrogenothermaceae* (*S. azorense*) still uses the two-step carboxylation of 2-oxoglutarate, suggesting that the two-step version of the rTCA cycle was present in the ancestor of both the *Aquificaceae* and the *Hydrogenothermaceae* (*Figure 9* and *Figure 10*). However, neither of the genes for the two-step citrate cleavage or two-step 2-oxoglutrate carboxylation is present in the *Desulfurobacteriaceae* (*Figure 9*). Finally, synteny analyses show that the four enzymes necessary to complement the one-step rTCA cycle variant in the *Desulfurobacteriaceae* from a theoretical ancestral linear rTCA are organized in a single operon (*Figure 8*).

Taking into consideration evidence from comparative genomics and phylogenetic analyses, the most parsimonious interpretation of the data suggests that the last common ancestor of the *Aquificae* had an incomplete form of the rTCA that did not proceed past the synthesis of 2-oxoglutarate, and that later on the cycle was closed following two independent evolutionary trajectories (*Figure 10*): (I) Gene duplication in the lineage that lead to the *Aquificaceae* and *Hydrogenothermaceae*; and (II) gene acquisition by horizontal gene transfer in the *Desulfurobacteriaceae* and *Hydrogenothermaceae* (the latter replaced the two-step version of the rTCA cycle with the one-step version) (*Figure 9* and *Figure 10*). *The reasons behind the presence of two distinct rTCA cycle variants within the extant Aquificae are not known. Braackman and Smith hypothesized that the one-step reactions might have evolved as a way to improve the thermodynamic efficiency of the rTCA cycle (Braakman and Smith, 2012). We hypothesize that temperature may also have played a role in preserving the ancient, more symmetric, two-step citrate cleavage rTCA cycle variant in Aquificaceae (Figure 10).* Members of this group have optimum growth temperatures (75–95°C) higher than those of the two other groups (60–75°C; *Table 1*), and the 'ancient' enzymes might be more stable at these high temperatures. In contrast, the 'newer' enzymes that catalyze the one-step citrate cleavage might have evolved to function optimally at lower temperatures.

Different members of the *Aquificae* use either one of the two versions of the rTCA cycle, and all the extant members of this phylum possess the genes encoding for the enzymes of the reductive acetyl-CoA pathway, with the exception of *codH* (encoding for the CO-dehydrogenase), which is only found in the *Desulfurobacteriaceae*. Therefore, we hypothesize that the last common ancestor of the *Aquificae* possessed the complete reductive acetyl-CoA pathway (*Figure 10*). While members of the *Desulfurobacteriaceae* kept the CodH due to their obligate anaerobic lifestyle, microaerophilic *Aquificae* (*Hydrogenothermaceae* and *Aquificaceae*) lost this extremely oxygen-sensitive enzyme (*Figures 4* and *10*). The presence of the gene encoding CodH in *P. marina* (*Hydrogenothermaceae*) is the only exception, and suggests that *codH* was lost independently in the two lineages (*Figure 10*).

Alltogether, our results suggest that the last common ancestor of the *Aquificae* combined the reductive acetyl-CoA pathway with an incomplete form of the rTCA that did not proceed past the synthesis of 2-oxoglutarate (*Figure 10*). A similar incomplete version of the rTCA pathway, consisting only of the reactions from acetyl-CoA to 2-oxoglutarate, is present in extant methanogens (*Berg, 2011*). Thus, our phylometabolic reconstruction of the ancestral state of carbon fixation in the

*Aquificae* (*Figure 10*) is conceptually consistent with chemoautotrophic processes of extant *Bacteria* and *Archaea*, and may represent the earliest carbon fixation pathway.

## Conclusions

We propose that the ancestor of *Thermovibrio ammonificans* was originally a hydrogen oxidizing, sulfur reducing bacterium that used a hybrid carbon fixation pathway for $CO_2$ fixation. The simultaneous presence of the rTCA cycle and of the reductive acetyl-CoA pathway of carbon fixation in *T. ammonificans* may represent a modern analog of the early carbon fixation phenotype, and suggests that the redundancy of central metabolic pathways was common in ancestral microorganisms. With the gradual rise of oxygen in the atmosphere, more efficient terminal electron acceptors became available and this lineage acquired genes that increased its metabolic flexibility - *e.g.,* the capacity to respire nitrate - along with enzymes involved in the detoxification of oxygen radicals. However, this lineage also retained its core, or ancestral, metabolic traits. Given the early branching nature of the phylum *Aquificae* and the ability of *T. ammonificans* and the *Desulfurobacteriaceae* to thrive in hydrothermal environments relying on energy sources of geothermal origins (namely carbon dioxide, hydrogen and elemental sulfur), we argue that these microorganisms represent excellent models to investigate how metabolism co-evolved with Earth's changing environmental conditions.

## Materials and methods

Strain isolation was described in (*Vetriani et al., 2004*). Growth condition, DNA extraction, sequencing strategy and automatic annotation were published in (*Giovannelli et al., 2012*).

### Manual curation of the genome

Manual curation of the genome was performed using blastn and blastp (*McGinnis and Madden, 2004*) against the non-redundant database (*Pruitt et al., 2007*), the conserved domain database (*Marchler-Bauer et al., 2005*), the Kyoto Encyclopedia of Genes and Genomes (*Kanehisa and Goto, 2000*) and the PFAM database (*Sonnhammer et al., 1998*). Coding sequence similarities were compared using translated protein sequence. Metabolic pathways were reconstructed on the basis of available genomic, physiologic and biochemical information and using Kyoto Encyclopedia of Genes and Genomes (*Kanehisa and Goto, 2000*) and SEED (*Overbeek et al., 2014*) pathways as template.

### Comparative genomics

Genome maps were drawn using Circos (*Krzywinski et al., 2009*), parsing blast results with *ad hoc* bash scripting. Comparative analyses between the genomes of *T. ammonificans*, those of representative members of the *Aquificae* (reported in *Table 3*), *Desulfurobacterium thermolithotrophum* (*Göker et al., 2011*) and *Caminibacter mediatlanticus* (*Giovannelli et al., 2011*) were performed using the GenomeEvolution pipeline CoGe (*Lyons et al., 2008*). *Aquificae* genomes were selected among all available genome within this phylum to maximize diversity while minimizing redundancy. All the available genomes of validly published *Aquificae* species were selected for analysis. To this set we added the reference genomes for the genera *Hydrogenivirga* and *Hydrogenobaculum*, as no genome sequence is available for validly published species of these genera. We also selected two additional genomes belonging to the genera *Desulfurobacterium* (the closest relative to the genus *Thermovibrio*) and *Persephonella*, respectively. Excluded genomes include either alternative assemblies of selected genomes or closely related genomes with a gapped genome similarity above 90%. LastZ pairwise alignments of selected genomes were used to draw three-way plots using the Hive Plot software (*Krzywinski et al., 2012*). Gene context and operonic structures were reconstructed using BioCyc (*Karp et al., 2005*) and FgenesB. Operons were manually screened and their structure selected based on gene context and available literature on the specific gene. When it was not possible to discriminate between the two alternative operonic predictions, both were reported in the text. Similarities between *T. ammonificans* genes and other prokaryotic genomes were found performing blastp analyses against the nr database. The top three blast results were retained and further analyzed, ranking the genes for their best hits. The procedure was repeated removing from the database the genome of *D. thermolithotrophum* and *Desulfurobacterium* sp. TC5-1, thus searching for the best hit outside of the *Desulfurobacteriaceae* family. The results were analyzed and

interactive Krona plots (*Ondov et al., 2011*) drawn linking the *T. ammonificans* genes with its closest match in the database. The interactive plots are accessible at DOI: 10.6084/m9.figshare.3178528. Images were drawn or edited using the open source vector drawing program Inkscape (http://inkscape.org/).

## Phylogenetic analyses

Phylogenetic analyses were performed using the approaches defined below for each tree. The phylogenetic tree presented in *Figure 1* was computed from the current 16S rRNA database alignment available from the ARB-SILVA project (http://www.arb-silva.de). The maximum likelihood three was computed from the ARB-SILVA alignment using PHYLML (*Guindon and Gascuel, 2003*) and the GTR model. The phylogenetic tree in *Figure 1—figure supplement 1* was constructed by aligning complete or near complete 16S rRNA sequences obtained from NCBI and representing the phylum *Aquificae*. The 16S rRNA sequences were aligned with ClustalO (*Thompson et al., 1997*) and Gblocks (*Castresana, 2000*) and the alignment was manually refined using SEAVIEW (*Galtier et al., 1996*). The maximum likelihood phylogeny was inferred from the alignment of 1455 sites using PHYML (*Guindon and Gascuel, 2003*), the GRT model and 1000 bootstrap replications. The CODH tree presented in *Figure 4* was computed using a selected set of amino acid sequences and the neighbor-joining method. Alignments were obtained using Muscle (*Edgar, 2004*) and Gblocks (*Castresana, 2000*), manually refined using SEAVIEW, and phylogenetic distances calculated using 255 sites and the Observed Divergence matrix. The neighbor-joining method was used to evaluate tree topologies using Phylo_win (*Galtier et al., 1996*) and their robustness was tested by bootstrap analysis with 1000 resamplings. The tree for the catalytic subunit of the [NiFe]-hydrogenases presented in *Figure 5* was computed using a selected set of amino acid sequences and the same approach described above for the CODH tree and was based on 539 sites. The accession number for the sequences used in trees presented in *Figure 1—figure supplement 1*, *Figures 4* and *5* are reported within each tree. The ATP citrate lyase phylogenetic tree was reconstructed with Bayesian inference and maximum likelihood methods. Both subunit of the ATP citrate lyase (AclB and AclA) were aligned individually to retrieved homologs using Muscle (*Edgar, 2004*) and SEAVIEW (*Galtier et al., 1996*). The alignments were concatenated and refined using Gblock (*Castresana, 2000*). A hypothetical ancestral ATP citrate lyase enzyme was manually constructed by concatenating the citryl-CoA synthetase and citryl-CoA lyase of *H. thermophilus* and *A. aeolicus*, respectively, and used as the outgroup (*Hügler et al., 2007*). The maximum likelihood phylogeny was inferred from the alignment using PHYML with the Wag substitution model (*Whelan and Goldman, 2001*) and 1000 bootstrap resamplings. The substitution model was selected based on AIC values using ProTest3 (*Darriba et al., 2011*). Bayesian phylogeny was inferred using MrBayes (*Ronquist and Huelsenbeck, 2003*) performing 500,000 generations with two parallel searches with the Wag amino acid matrix model (selected using forward selection among all possible substitution models) and a burn-in of 125,000 generations. Both tree were computed on 1035 identified sites. Accession numbers for the tree presented are reported in *Figure 9—source data 1*. A combination of phylogenetic analysis and comparative genomic analyses were used to identify lateral gene transfer events.

## Phylometabolic reconstruction of the carbon fixation pathway within the *aquificae*

Phylometabolic analysis (*Braakman and Smith, 2012*) was used to investigate the carbon fixation pathway in *T. ammonificans* and its evolutionary relationship to the carbon fixation pathways present in other members of the *Aquificae* phylum. In phylometabolic analyses, the metabolic pathways of the organism under investigation are compared to those found in related organisms both within and across neighboring clades. By focusing on the pathways, the comparison may reveal variations in multi-enzyme functional units, providing context for the completion of the pathway within the networks of individual organism, while also allowing for the identification of ancestral states and horizontal gene transfer events. The resulting phylometabolic tree includes multiple complete pathways to common essential metabolites, and suggests which evolutionary substitutions are allowed (at either organism or ecosystem levels) among these pathways (see [*Braakman and Smith, 2012*] and reference therein for a more extensive description of the principles underlying this approach). We reconstructed the carbon fixation pathways in representative genomes of the *Aquificae*, and

compared them. Information regarding the carbon fixation metabolic network was implemented using phylogenetic and comparative genomic information to help reconstruct possible ancestral states of the carbon fixation network based on maximum parsimony principles.

## Protein extraction, digestion and identification

*T. ammonificans* cell pellets were washed in TE buffer (10 mM Tris-HCl pH 7.5, 10 mM EDTA pH 8.0, containing Roche cOmplete protease inhibitor) and soluble proteins were extracted as described by (*Heinz et al., 2012*). Briefly, cells were disrupted by sonication (2 × 25 s), cell debris was pelleted and protein concentrations in the supernatant were determined according to (*Bradford, 1976*). 20 μg of protein extract were loaded onto a precast 10% polyacrylamide mini gel in technical triplicates for 1D PAGE (150 V, 45 min). After staining with Coomassie Brilliant Blue, protein-containing gel lanes were excised and cut into 10 equal subsamples each, which were destained (200 mM $NH_4HCO_3$, 30% acetonitrile) and digested with trypsin solution (1 μg/ml, Promega, Madison WI, USA) at 37°C over night, before peptides were eluted from the gel pieces in an ultrasonic bath (15 min). As described by (*Xing et al., 2015*), peptide mixes were subjected to reversed phase C18 column chromatography on a nano-ACQUITY-UPLC (Waters Corporation, Milford, MA, USA). Mass spectrometry (MS) and MS/MS data were recorded with an online-coupled LTQ-Orbitrap mass spectrometer (Thermo Fisher Scientific Inc., Waltham, MA, USA). MS data were searched against the forward-decoy *T. ammonificans* protein database using Sequest (Thermo Fisher Scientific, San Jose, CA, USA; version 27, revision 11) and identifications were filtered and validated in Scaffold (http://www.proteomesoftware.com), as described previously (*Heinz et al., 2012*). Data for all three technical replicates were merged and exclusive unique peptide count values of all proteins were exported for calculation of normalized spectral abundance factors (NSAF), which are given in *Figure 3—source data 1* in %, *i.e.*, as relative abundance of each protein in % of all identified proteins.

## Acknowledgements

The authors gratefully acknowledge the support of the Deep Carbon Observatory. Thanks to Sandra Beyer for assistance in the lab and to Sebastian Grund for MS measurements. The genome of *Thermovibrio ammonificans* was sequenced under the auspices of the US Department of Energy. Work on *T. ammonificans* was supported, in part, by NSF grants MCB 04–56676, OCE 03–27353, MCB 08–43678, OCE 09–37371, OCE 11–24141, OCE11-36451, and NASA grant NNX15AM18G to CV, NSF grant MCB 15–17567 to CV and DG, and by the New Jersey Agricultural Experiment Station. DG was supported by a Postdoctoral Fellowship from the Institute of Marine and Coastal Sciences and a Postdoctoral Fellowship from the Center for Dark Energy Biosphere Investigations (C-DEBI). Funding to SMS was provided by NSF grant OCE-1136727 and a senior fellowship by the Alfried Krupp Wissenschaftskolleg Greifswald, Germany. This publication was in part supported by the ELSI Origins Network (EON), which is supported by a grant from the John Templeton Foundation. The opinions expressed in this publication are those of the authors and do not necessarily reflect the views of the John Templeton Foundation. This paper is C-DEBI contribution 368.

## Additional information

### Funding

| Funder | Grant reference number | Author |
| --- | --- | --- |
| National Science Foundation | MCB 04-56676 | Costantino Vetriani |
| National Aeronautics and Space Administration | NNX15AM18G | Costantino Vetriani |
| National Science Foundation | OCE 03-27353 | Costantino Vetriani |
| National Science Foundation | MCB 08-43678 | Costantino Vetriani |
| National Science Foundation | OCE 09-37371 | Costantino Vetriani |
| National Science Foundation | OCE 11-24141 | Costantino Vetriani |
| National Science Foundation | MCB 15-17567 | Donato Giovannelli |

| | | Costantino Vetriani |
| National Science Foundation | OCE-1136727 | Stefan M Sievert |

The funders had no role in study design, data collection and interpretation, or the decision to submit the work for publication.

## Author contributions

DG, Conceptualization, Data curation, Formal analysis, Investigation, Methodology, Writing—original draft, Writing—review and editing; SMS, Conceptualization, Formal analysis, Funding acquisition, Methodology, Writing—review and editing; MH, Conceptualization, Formal analysis, Investigation, Writing—review and editing; SM, Data curation, Formal analysis, Methodology, Writing—review and editing; DB, Formal analysis, Methodology, Writing—review and editing; TS, Conceptualization, Data curation, Methodology, Writing—review and editing; CV, Conceptualization, Formal analysis, Supervision, Funding acquisition, Writing—original draft, Project administration, Writing—review and editing

## Author ORCIDs

Donato Giovannelli, http://orcid.org/0000-0001-7182-8233
Michael Hügler, http://orcid.org/0000-0002-2820-0333
Costantino Vetriani, http://orcid.org/0000-0002-8141-8438

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
