## [Decision Letter]

Thank you for submitting your article "Insight into the evolution of microbial metabolism from the deep-branching bacterium, *Thermovibrio ammonificans*" for consideration by *eLife*. Your article has been reviewed by three peer reviewers, and the evaluation has been overseen by a Reviewing Editor and Richard Losick as the Senior Editor. The reviewers have opted to remain anonymous.

The reviewers have discussed the reviews with one another and the Reviewing Editor has drafted this decision to help you prepare a revised submission.

Following below please find the assessments of the three external reviews. As you will see, the paper was overall quite positively received by the reviewers, but there is some substantial additional work in the data analyses and presentation of the paper that needs to be done before a substantially revised submission can be considered for publication in *eLife*.

*Reviewer #1:*

The authors use a comparative genomic approach associated to a proteomic analysis to reconstruct the metabolism of *Thermovibrio ammonificans* and draw an interesting picture of its possible evolution. Particularly interesting is the observation that the genome of *T. ammonificans* codes for enzymes putatively involved in the reductive acetyl-CoA pathway and in the reductive TCA (rTCA) cycle.

*T. ammonificans* encodes an unusual CodH (type V), whose function is still unknown. Based on the assumption that the type V CodH works in the rTCA cycle, the authors suggest that in *T. ammonificans* the reductive acetyl-CoA pathway and the rTCA cycle are both operational and that the bacterium uses one or the other way to fix carbon based on the environmental conditions.

This is a key point of the manuscript and should have some experimental support.

The manuscript is well presented although in some parts the reader needs to have high familiarity with microbial evolution and metabolism.

I have some questions and concerns:

1) Subsection “Carbon Fixation”, second paragraph: why it is not clearly stated the variant used by *T. ammonificans*?

2) Figures are not numbered in the order of citation in the text. For example, Figure 7 and Figure 8 (subsection “Carbon Fixation”, last paragraph and subsection “Hydrogen oxidation”, second paragraph, respectively) appear in the text right after Figure 3 while Figure 4 is only cited in the third paragraph of the subsection “Insight into the evolution of carbon fixation”. Also Figure 5 appears before Figure 4 (subsection “Comparative genomic of *T. ammonificans*: ancestral and acquired metabolic traits”, second paragraph)

3) Subsection “Carbon Fixation”, last paragraph: the authors refer to two genes coding for CodH. But Theam_0999 is not listed in Table 2 and it is not clearly explained what it is. Only in the aforementioned paragraph it is stated that it has 81% similarity with a gene of *D. thermolithotrophum*. Theam_0999 is also not listed in Table 3, then it is presumably not expressed in the tested growth conditions.

4) Subsection “Carbon Fixation”, last paragraph: what are the bases for the hypothesis that type V CodH may represent the most ancient version of these enzymes?

5) Subsection “Sulfur reduction”: if Theam_1321 has no homologs in the database how was its putative function (NAD- or FAD-dependent reductase) assigned?

6) Subsection “Activated methyl cycle”: I missed what is the evidence to conclude that "In *T. ammonificans*, the activated methyl cycle is of the second type."

*Reviewer #2:*

Giovanelli et al. performed comparative genomic, proteomic and phylometabolic analyses, of the anaerobic thermophile, *Thermovibrio ammonificans* and identified two distinct groups of genes. One that codes for enzymes that do not require oxygen and use substrates of geothermal origin and another that may reflect the rise of oxygen on Earth. Based on this they propose the last common ancestor was a hydrogen oxidizing, sulfur reducing bacterium that used a hybrid pathway for CO2 fixation. I find this an interesting manuscript, with potential interest, but have concerns regarding the methods used in the phylogenomic analyses and also the general scope of the paper.

First, the authors start out in the Abstract writing about early earth and evolution and origins of life, but the paper itself is more of a descriptive narrative of the different pathways encoded and expressed by the organism. The links to evolution and origin of life and this organism are buried in the paper behind long stretches of descriptive text that reads more like a genome announcement paper than a manuscript on the evolution of the group. Second, the authors conclusion that *Thermovibrio ammonificans* encodes ancestral pathways (e.g., hydrogen oxidation) and more recently acquired ones (e.g., nitrate reduction) and a hybrid pathway for CO2 fixation are not novel conclusions as they have already been shown in several of the authors prior works.

The main part of this manuscript, that has potential as a novel contribution, is the genome evolution aspect. However, the authors do not use a strong phlyogenomic approach to support their claim of the original metabolism of the common ancestor. From what I can tell, they base this conclusion on a comparison of the target organism to only two other related strains (in the Methods section “Comparative genomics”). The proper way to test such a hypothesis would be to take all available genomes from the two clades in question and perform a phylogenomic analysis across all members to identify those genes that have been vertically inherited (e.g., shared protein clusters) as opposed to horizontally inherited (Enright et al., 2002 NAS). This can be done using a Markov Cluster algorithm for example (Nelson-Sathi et al., 2015 Nature). Vertically inherited genes can be defined based on two simple criteria: (1) the protein should be present in at least two higher taxa of the two clades in question and (2) its tree should recover monophyly of the two clades (Weiss et al. 2016 Nature Microbiology). Genes meeting both criteria are unlikely to have undergone LGT, and thus were probably present in the last common ancestor. Unfortunately, such an analysis is missing in the authors study, but is needed to support this main conclusion. I cannot support publication until this major concern is addressed.

Specific comments:

Introduction, second paragraph: There is a lot of evidence suggesting that the reductive acetyl CoA was the first carbon fixation pathway, and that the first organism was actually a methanogen, not a bacterium, and that this organism evolved under lower temperature hydrothermal conditions. Please read some of the recent papers from Bill Martins group, for example Weiss et al. 2016 (Nature Microbiology), where they did phylogenomic analyses of all prokaryotic genomes to date.

Introduction, third paragraph: Regarding the reductive acetyl CoA pathway and the origins of life I suggest you should also read the new review by Sojo et al. in Astrobiology (2016).

Introduction, third paragraph, last sentence: This is not a universally accepted concept as the authors imply with this sentence. Please see Weiss et al. 2016 (Nature Microbiology), who show that LUCA was likely a methanogen.

Subsection “Carbon fixation”, last paragraph: If not all the genes in the pathway are encoded, then how do you explain your discovery of all the proteins (stated a few paragraphs above)?

Subsection “Carbon fixation”, last paragraph: Please provide citations for the presence of this pathway in sulfate reducers and anammox bacteria.

“We hypothesize that the type V CodH may represent the most ancestral version of this class of enzymes, and may catalyze the reduction of CO_2_ to CO”: Do you base this hypothesis on your own data or someone else?

Subsection “Hydrogen oxidation”, end of third paragraph: How is this link suggested here? This is not clearly explained.

Subsection “Comparative genomic of *T. ammonificans*: ancestral and acquired metabolic traits”, second paragraph: Why did you choose specifically these organisms to compare?

Subsection “Comparative genomic of *T. ammonificans*: ancestral and acquired metabolic traits”, second paragraph: From what you write here it is not clear how your synteny analysis shows that this horizontal gene transfer occurred.

Subsection “Comparative genomic of *T. ammonificans*: ancestral and acquired metabolic traits”, third paragraph: Can you discuss the connections to the Weiss et al. paper (2016) which has already analyzed this topic in more greater detail?

Subsection “Comparative genomic of *T. ammonificans*: ancestral and acquired metabolic traits”, end of third paragraph: It is already well known that reductive acetyl CoA pathway emerged prior to photosynthesis. What do you mean that the central metabolism of *T. ammonificans* is "ancestral"? This is very general, and not a new idea.

Subsection “Comparative genomic of *T. ammonificans*: ancestral and acquired metabolic traits”, last two paragraphs: This is an interesting idea, but where is your data supporting this hypothesis? No figures or tables are cited in the text…

Subsection “Insight into the evolution of carbon fixation”, first paragraph: The Brakeman and Smith study is outdated, more updated is Weiss et al. 2016 Nature Microbiology.

Subsection “Insight into the evolution of carbon fixation”, end of first paragraph: I disagree, new analyses of all prokaryotic genomes and biochemical pathways looking at the oldest conserved genes between all organisms suggest LUCA was a methanogen with reductive acetyl CoA pathway (Weiss et al. 2016). Not *Aquificae* as the authors propose here.

*Reviewer #3:*

This is a very comprehensive and beautifully written genomic & proteomic analysis of *Thermovibrio ammonificans*, followed by a comparative analysis of its C fixation pathways with those of the *Aquificaceae* and the sulfur/hydrogen-oxidizing *epsilonproteobacteria*. On first reading, I can only find minor details that need attention (see below). After seeing how cursory many genomes are treated, it was a pleasure to read this analysis and its careful exploration of autotrophic pathway evolution.

The underpinnings of this manuscript – dividing biochemical pathways into ancestral and modern, and then dissecting the genome of a bacterium into ancient and modern layers, like different time horizons in an archaeological excavation – require a short comment. The complexity of even "ancient" pathways, genes and enzymes reflects the fact that every "ancient" gene and enzyme that still exists today has a billion-year history of evolutionary finetuning and, often enough, lateral gene transfer and functional repurposing. Nothing has survived from the dawn of life unchanged. This short preface should explain why I take some introductory statements of this article with a grain of salt, for example the big jump from Wachterhauser to the rTCA cycle (see below).

Introduction, second paragraph: If this assessment – the *Aquificae* as an early lineage – is based on 16S rRNA gene sequences, you should say this explicitly.

Introduction, third paragraph: I don't have access to Waechtershauser 1988 right now, but his article is focusing on surface catalysis of CO_2_, CO and small organic molecules being assembled into larger molecules, all on a metal-sulfide or pyrite surface; the theme is prebiotic "autotrophic" syntheses catalyzed by a reactive surface. The fully developed rTCA cycle is far away from that.

The last sentence of the subsection “Conclusions” – "we argue that proposed model for the evolution of metabolism presented in this paper may represent the blueprint of the chemistry of early life" – goes too far in the same direction. Most problematically in origin of life scenarios, early probiotic reactions, like covalent bond formations at reactive interfaces or charge separation at mineral surfaces, are separated by a huge gulf from fully developed enzymatic systems, encoded by genes, transcribed by mRNA, translated by ribosomes… you get the idea. Even ancestral enzymes and pathways only provide a view on the evolution of fully differentiated, metabolically active and self-sustaining microorganisms that we would recognize today [with proper genomes, not to forget], but they do not allow a view to the really early stages and origin of life; theories on communal genomes before cell differentiation, and early evolution of protein domains and structural subunits try to fill the gap. This difference should be acknowledged. Again, nothing has survived from the dawn of life unchanged.

[Editors' note: further revisions were requested prior to acceptance, as described below.]

Thank you for resubmitting your work entitled "Insight into the evolution of microbial metabolism from the deep-branching bacterium, *Thermovibrio ammonificans*" for further consideration at *eLife*. Your revised article has been reevaluated by Gisela Storz (Senior Editor), a Reviewing Editor, and two reviewers.

One of the reviews revealed some considerable major concerns. Normally, this would be reason for rejection of the paper. However, some of the issues may be based on a misunderstanding with the original reviews. Therefore, if you can address the remaining major concerns we would consider a further revised submission. The critical review comments follow below.

Following are the key issues that need to be addressed:

Much of the rebuttal consists of straw man arguments regarding the Weiss et al. study and origins of life issue that detract from my main points raised about their analyses. I apologize if the authors got the wrong idea from my original review, which was not to ask them to re-write their manuscript on the origins of life. I was not a co-author on that paper, but it is a good example of extensive taxon sampling and (in principle) a good method for determining vertical inheritance. As the authors note in their rebuttal whether or not Weiss et al. actually followed their method is currently debated in online comment forums, but this is outside the scope of the authors’ manuscript. By focusing on this, I think that the authors may have missed the main point of my review, which was that their methods for phylogenomic analyses were not sufficient (or at least not sufficiently explained in the Methods). It is possible that analyses were done satisfactorily but this is impossible to tell from the information provided. I have re-read the authors Methods carefully a second time in the revised manuscript, yet it is not possible to tell how the authors processed their data in the different analyses.

My main point is that the authors need to explain in their Methods exactly how they did the different phylogenetic analyses, and provide a satisfactory justification for why they chose to treat their data in the way they did (e.g., why no concatenated gene phylogenies?) instead of following other conventional methods for interpreting bacterial phylogenomics. For example, concatenated gene trees possess more phylogenetic signals, and are less susceptible to the stochastic errors than those built from single genes (Wu and Eisen, 2008; Jeffroy et al. 2006). From what I can tell the authors performed single gene phylogenies of a handful of genes picked from the genomes.

The authors write in their rebuttal: "That is incorrect, as members of the *Desulfurobacteriaceae, Aquificaceae* and *Epsilonproteobacteria* were included in the analyses and discussed throughout the manuscript. We used all available *Aquificae* genomes (as indicated in Table 3), and further focused only on the closest relative and the *Epsilonproteobacteria* as an ecological proxy."

A quick search of all available genomes from the *Aquificae* in IMG reveals that there are genomes from 29 representatives available for analysis. Table 3 in the manuscript contains 17 of these, which cover the major clades but does not include all available genomes from each clade. Thus, the statement that the authors make that "Comparative analyses between the genomes of *T. ammonificans*, all available *Aquificae* genomes…" is incorrect. Taxon sampling (e.g., n=17 vs. n=29) can have an impact on the results of phylogenetic analyses. Usually the addition of more representatives can provide a better resolution. Please justify why you selected only these 17 for analysis.

In the Methods section "Comparative genomics": It is clear from this that you used the genomes in Table 3. But, it is not clear whether these same genomes were used to create the phylogenetic analyses or not.

In the Methods section "Phylogenetic Methods": The authors describe methods for 16S phylogenies and ATP citrate lyase phylogenies. No information is provided regarding the Ni-Fe hydrogenase phylogeny (Figure 5), CO dehydrogenases (Figure 4), or any other phylogenetic analyses that the authors base their conclusions on. At least, not from what I can tell after reading the section. I do not know whether this is intended or a mistake. Since the authors base their conclusions based on these results it seems like a good idea to explain how these analyses were done. For example, please state number of aligned resides for each tree, whether it is amino acids of DNA data, e.g. the standard information that readers deserve to know so that the authors work can be openly evaluated by the scientific community after publication. Another major question remains, which is why have the authors performed only single gene phylogenies as opposed to concatenated gene phylogenies? Please explain. Concatenated gene phylogenies are less susceptible to stochastic errors than those built from single genes.

---

## [Author Response]

*Reviewer #1:*

*The authors use a comparative genomic approach associated to a proteomic analysis to reconstruct the metabolism of Thermovibrio ammonificans and draw an interesting picture of its possible evolution. Particularly interesting is the observation that the genome of T. ammonificans codes for enzymes putatively involved in the reductive acetyl-CoA pathway and in the reductive TCA (rTCA) cycle.*

*T. ammonificans encodes an unusual CodH (type V), whose function is still unknown. Based on the assumption that the type V CodH works in the rTCA cycle, the authors suggest that in T. ammonificans the reductive acetyl-CoA pathway and the rTCA cycle are both operational and that the bacterium uses one or the other way to fix carbon based on the environmental conditions. This is a key point of the manuscript and should have some experimental support.*

Comment on the functionality of the reductive Acetyl-CoA pathway. We agree that the role of the reductive Acetyl-CoA pathway and of the CodH in *T. ammonificans* is controversial. To begin to understand the role of the reductive Acetyl-CoA pathway in *T. ammonificans* we carried out experiments aimed to assess the C isotopic fractionation during autotrophic growth of the bacterium under different electron accepting conditions (nitrate and sulfur). Our hypothesis was that under less efficient electron accepting conditions (i.e., sulfur reduction), *T. ammonificans* might preferentially use the reductive acetyl-CoA pathway, which requires ~1 ATP *vs* 2-3 ATP for the rTCA cycle. Briefly, we analyzed the biomass ^13^C fractionation with respect to the CO2 used as substrate by the organisms in triplicated independent experiments. *T. ammonificans* biomass δ 13 C isotopic fractionation ranged between -6 ‰ and -8 ‰. The values obtained are in the range of values reported in the literature for the rTCA cycle (Berg et al. 2010), confirming the activity of rTCA as main strategy of carbon fixation under the condition tested. Cells of *T. ammonificans* grown on sulfur as terminal electron acceptor consistently had isotopic value more negative than population grown in nitrate reducing condition of about 2 ‰. While this difference is small and does not support the activity of the reductive acetyl-CoA pathway, it raises interesting questions regarding the differential isotopic fractionation of carbon while growing with alternative electron acceptors. Because results were inconclusive we decided against including these new experiments in our manuscript. Further experiments that go beyond the scope of this manuscript, whose original objective was to put forth a number of hypotheses based on genomic reconstruction, will need to be designed to shed light on the function of the reductive Acetyl-CoA pathway in *T. ammonificans*.

*The manuscript is well presented although in some parts the reader needs to have high familiarity with microbial evolution and metabolism.*

*I have some questions and concerns:*

*1) Subsection “Carbon Fixation”, second paragraph: why it is not clearly stated the variant used by T. ammonificans?*

A sentence stating that the asymmetric variant of the rTCA cycle is found in *T. ammonificans* was added (subsection “Carbon Fixation”, second paragraph).

*2) Figures are not numbered in the order of citation in the text. For example, Figure 7 and Figure 8 (subsection “Carbon Fixation”, last paragraph and subsection “Hydrogen oxidation”, second paragraph, respectively) appear in the text right after Figure 3 while Figure 4 is only cited in the third paragraph of the subsection “Insight into the evolution of carbon fixation”. Also Figure 5 appears before Figure 4 (subsection “Comparative genomic of T. ammonificans: ancestral and acquired metabolic traits”, second paragraph)*

The figure order and numbering has been corrected throughout the manuscript. Thank you for pointing out this inconsistency!

*3) Subsection “Carbon Fixation”, last paragraph: the authors refer to two genes coding for CodH. But Theam_0999 is not listed in Table 2 and it is not clearly explained what it is. Only in the aforementioned paragraph it is stated that it has 81% similarity with a gene of D. thermolithotrophum. Theam_0999 is also not listed in Table 3, then it is presumably not expressed in the tested growth conditions.*

Theam_0999 codes for an iron-sulfur cluster accessory protein to CodH (coded by the gene Theam_1337). Reference to Theam_0999 is now present only in the last paragraph of the subsection “Carbon Fixation”.

*4) Subsection “Carbon Fixation”, last paragraph: what are the bases for the hypothesis that type V CodH may represent the most ancient version of these enzymes?*

The possibilty that type V CodH constitute the ancestral form of the enzyme was suggested by Frank Robb while discussing the Techtmann et al., 2012 paper. Briefly, the structure of the protein retains all the catalytic and binding residuals of a canonical CodH while presenting a simpler (and smaller) overall structure. A clarifying sentence was added to the last paragraph of the subsection “Carbon Fixation”.

*5) Subsection “Sulfur reduction”: if Theam_1321 has no homologs in the database how was its putative function (NAD- or FAD-dependent reductase) assigned?*

Subsection “Sulfur reduction”: The statement of Theam_1321 being unique to *T. ammonificans* was revised.

*6) Subsection “Activated methyl cycle”: I missed what is the evidence to conclude that "In T. ammonificans, the activated methyl cycle is of the second type."*

Comment on the activated methyl cycle: In *T. ammonificans* the activated methyl cycle involves the *sahH* gene, while a *luxS* homolog is absent. We clarified this in the subsection “Activated methyl cycle”. A figure depicting the cycle described in the text was also added a diagram of the activated methyl cycle in *T. ammonificans* as Figure 3—figure supplement 2.

*Reviewer #2:*

*Giovanelli et al. performed comparative genomic, proteomic and phylometabolic analyses, of the anaerobic thermophile, Thermovibrio ammonificans and identified two distinct groups of genes. One that codes for enzymes that do not require oxygen and use substrates of geothermal origin and another that may reflect the rise of oxygen on Earth. Based on this they propose the last common ancestor was a hydrogen oxidizing, sulfur reducing bacterium that used a hybrid pathway for CO2 fixation. I find this an interesting manuscript, with potential interest, but have concerns regarding the methods used in the phylogenomic analyses and also the general scope of the paper.*

*First, the authors start out in the Abstract writing about early earth and evolution and origins of life, but the paper itself is more of a descriptive narrative of the different pathways encoded and expressed by the organism. The links to evolution and origin of life and this organism are buried in the paper behind long stretches of descriptive text that reads more like a genome announcement paper than a manuscript on the evolution of the group.*

The first general comment was about the evolution of early metabolism, which is the main focus of the manuscript. Reviewer 2 stated that the manuscript was too focused on the central metabolism of *T. ammonificans*, and that such focus is a distraction from the evolution theme. Our answer to that comment is that, to draw *any* general conclusion on the evolution of metabolism in the *Desufurobacteraceae*, it was critical to investigate in depth the central metabolic pathways in our model organism. We tried to strike a balance between the two topics by reporting our genome-derived findings on the central metabolic pathways in *T. ammonificans* (i.e., our data) in the Results section, while discussing the evolutionary implications of our findings in the Discussion section. Reviewers 1 and 3 praised the presentation style of the manuscript.

*Second, the authors conclusion that Thermovibrio ammonificans encodes ancestral pathways (e.g., hydrogen oxidation) and more recently acquired ones (e.g., nitrate reduction) and a hybrid pathway for CO2 fixation are not novel conclusions as they have already been shown in several of the authors prior works.*

The second general comment of reviewer 2 stated that our conclusions on the ancestral and acquired genome of *T. ammonificans*, including the hybrid carbon fixation pathway, are not novel and that we had shown these data previously. That is absolutely incorrect, as the data presented here, along with our interpretation, is original work never published before. We did not publish other papers linking the evolution and metabolism of the *Desulfurobacteraceae* family and, to our knowledge, neither did others.

*The main part of this manuscript, that has potential as a novel contribution, is the genome evolution aspect. However, the authors do not use a strong phlyogenomic approach to support their claim of the original metabolism of the common ancestor. From what I can tell, they base this conclusion on a comparison of the target organism to only two other related strains (in the Methods section “Comparative genomics”). The proper way to test such a hypothesis would be to take all available genomes from the two clades in question and perform a phylogenomic analysis across all members to identify those genes that have been vertically inherited (e.g., shared protein clusters) as opposed to horizontally inherited (Enright et al., 2002 NAS). This can be done using a Markov Cluster algorithm for example (Nelson-Sathi et al., 2015 Nature). Vertically inherited genes can be defined based on two simple criteria: (1) the protein should be present in at least two higher taxa of the two clades in question and (2) its tree should recover monophyly of the two clades (Weiss et al. 2016 Nature Microbiology). Genes meeting both criteria are unlikely to have undergone LGT, and thus were probably present in the last common ancestor. Unfortunately, such an analysis is missing in the authors study, but is needed to support this main conclusion. I cannot support publication until this major concern is addressed.*

The third general comment was related to our comparative genomics and phylometabolic analyses. Reviewer 2 states that the genome of *T. ammonificans* was compared only to two closely related species. That is incorrect, as members of the *Desulfurobacteriaceae, Aquificaceae* and *Epsilonproteobacteria* were included in the analyses and discussed throughout the manuscript. We used all available *Aquificae* genomes (as indicated in Table 3), and further focused only on the closest relative and the *Epsilonproteobacteria* as an ecological proxy.

*Specific comments:*

*Introduction, second paragraph: There is a lot of evidence suggesting that the reductive acetyl CoA was the first carbon fixation pathway, and that the first organism was actually a methanogen, not a bacterium, and that this organism evolved under lower temperature hydrothermal conditions. Please read some of the recent papers from Bill Martins group, for example Weiss et al. 2016 (Nature Microbiology), where they did phylogenomic analyses of all prokaryotic genomes to date.*

Comments on the “first organism being a methanogen”. Reviewer 2 refers to Weiss et al., 2016 (DOI: 10.1038/NMICROBIOL.2016.116), which was published *after* our manuscript was submitted to *eLife*. The main message of the Weiss et al. paper is that the reconstruction of the most ancestral microbial genome (the authors refer to the genome of LUCA – Last Universal Common Ancestor) suggests that the earliest microorganisms were acetogens (bacteria) and/or methanogens (archaea). Based on their genome reconstruction the authors conclude that LUCA was an anaerobic, thermophilic autotroph that fixed CO2 via the reductive Acetyl-CoA pathway and could live on H2, CO2 and N2 in an environment similar to deep-sea hydrothermal vents. Hence, the statement from reviewer 2 that Weiss et al. concluded that LUCA was a methanogen is incorrect, as it could have been an acetogen as well. That LUCA might have been an acetogen is in agreement with the hypothesis put forward by Braakman and Smith, 2012, who took a completely different approach to reconstruct the origin of microbial metabolism. Having said that, we tend to agree with that scenario proposed by Weiss et al. (and by Braakman and Smith), which, with the exception of nitrogen fixation, fits very nicely with the metabolism of *T. ammonificans*. It needs to be noted that the presence of a nitrogenase-encoding gene in the genome of LUCA proposed by Weiss et al. is currently a subject of debate on PubMed Commons(https://www.ncbi.nlm.nih.gov/pubmed/27562259). However, while basically our findings are in agreement with the Weiss et al. paper, we want to stress that our study was not designed to identify the earliest form of microbial metabolism. We demonstrate that *T. ammonificans* is a good model to reconstruct the evolutionary history of early metabolism but we did not suggest that the lineage that led to the present-day *Desulfurobacteriaceae* and to *T. ammonificans* had the same metabolism of LUCA. What happened prior to the emergence of the *Desulfurobacteriaceae* lineage cannot be gleaned by our analyses, and it is not an objective of our study.

*Introduction, third paragraph: Regarding the reductive acetyl CoA pathway and the origins of life I suggest you should also read the new review by Sojo et al. in Astrobiology (2016).*

Comment on the reductive Acetyl-CoA pathway and the origin of life. Reviewer 2 refers to the recent work by Sojo et al., 2015 (DOI: 10.1089/ast.2015.1406), which proposes three interesting – but so far untested – hypotheses for CO2 reduction in a very specific environment – alkaline vents. The first hypothesis is that the synthesis of Acetyl-CoA was originally driven by the geochemistry of alkaline vents and it was not genetically encoded. The second hypothesis is that the synthesis of Acetyl-CoA as we know it in extant methanogens and acetogens was predated by a “lost” pathway not found in modern cells. The authors themselves refer to this hypothesis as quite “radical” as, among other assumptions, requires that the Haedean oceans were fairly oxidizing. The third hypothesis implies that the redox gradient at alkaline vents was sufficient to reduce CO2. Hypotheses 1 and 3 rely essentially on prebiotic chemistry, while hypothesis 2 assumes a “lost” pathway of which there is no trace in extant microorganisms. We apologize if we conveyed the wrong message, but our manuscript is not about prebiotic chemistry and the origin of life. Our manuscript is about the reconstruction of the central metabolic pathways in an early-branching – but modern – bacterium, *T. ammonificans*. From that reconstruction we made an effort to tease apart “early” and “acquired” traits to reconstruct the evolution of metabolism in this bacterium. But by “early” or “ancestral” we do not imply “at the origin of life”. Again, what happened prior to the emergence of the *Desulfurobacteriaceae* lineage cannot be gleaned by our analyses, and it is not an objective of our study. By stating that, among the six known carbon fixation pathways, the rTCA cycle and the reductive Acetyl-CoA pathway are “good candidates for the ancestral carbon fixation pathway”, we refer to a fairly extensive body of work published by others (including work by Bill Martin’s group, to whom reviewer 2 often refers) and we mean “good candidates” – we do not imply that such pathways *are* the original pathways.

*Introduction, third paragraph, last sentence: This is not a universally accepted concept as the authors imply with this sentence. Please see Weiss et al. 2016 (Nature Microbiology), who show that LUCA was likely a methanogen.*

See our first response to specific comments.

*Subsection “Carbon fixation”, last paragraph: If not all the genes in the pathway are encoded, then how do you explain your discovery of all the proteins (stated a few paragraphs above)?*

Comment on the discrepancy between gene and protein detection. We changed “All” to “Most” (subsection “The proteome of *T. ammonificans*”).

*Subsection “Carbon fixation”, last paragraph: Please provide citations for the presence of this pathway in sulfate reducers and anammox bacteria.*

Request to add a reference to substantiate the claim that the CodH is also found in sulfate reducers and anammox. The Berg et al., 2010 reference was added to the last paragraph of the subsection “Carbon Fixation”.

*“We hypothesize that the type V CodH may represent the most ancestral version of this class of enzymes, and may catalyze the reduction of CO_2_ to CO”: Do you base this hypothesis on your own data or someone else?*

Comment of the hypothesis that type V CodH is the most ancestral group. Please see related answer to reviewer 1.

*Subsection “Hydrogen oxidation”, end of third paragraph: How is this link suggested here? This is not clearly explained.*

Comment on the link between the distribution of hydrogenases and oxygen adaptation. Type 2 hydrogenases are found in microaerobic members of the *Aquificacea* and *Hydrothermaceae*, while type 3a hydrogenases are found in the strictly anaerobic *Desulfurobacteriaceae* and in methanoges. We clarified this statement in the text (subsection “Hydrogen oxidation”, third paragraph).

*Subsection “Comparative genomic of T. ammonificans: ancestral and acquired metabolic traits”, second paragraph: Why did you choose specifically these organisms to compare?*

Subsection “Comparative genomic of *T. ammonificans*: ancestral and acquired metabolic traits”, second paragraph: Comment on the choice of organisms for direct comparative genomic analyses. As explained in the text, we choose *D. thermolithotrophum* because it is the closest relative to *T. ammonificans* whose genome is available, and *C. mediatlanticus* because, while being phylogetically distant from *T. ammonificans*, occupies a similar ecological niche and has a similar central metabolism (it is a hydrogenotroph that conserves energy by nitrate or sulfur reduction and fixes CO2 via the rTCA cycle).

*Subsection “Comparative genomic of T. ammonificans: ancestral and acquired metabolic traits”, second paragraph: From what you write here it is not clear how your synteny analysis shows that this horizontal gene transfer occurred.*

Comment on the possible horizontal gene transfer (HGT) of the ATP citrate lyase between *Epsilonproteobacteria* and *Desulfurobacteraceae*. This HGT event was previously suggested by Hügler et al., 2007, Env. Microbiol 9:81. However, we agree with reviewer 2 that the synteny diagram in Figure 8 does not indicate such HGT event, and we removed the last sentence of the paragraph.

*Subsection “Comparative genomic of T. ammonificans: ancestral and acquired metabolic traits”, third paragraph: Can you discuss the connections to the Weiss et al. paper (2016) which has already analyzed this topic in more greater detail?*

*Subsection “Comparative genomic of T. ammonificans: ancestral and acquired metabolic traits”, end of third paragraph: It is already well known that reductive acetyl CoA pathway emerged prior to photosynthesis. What do you mean that the central metabolism of T. ammonificans is "ancestral"? This is very general, and not a new idea.*

Comments on relating our analysis of the genome of *T. ammonificans* with the work of Weiss et al., 2016, and on the ancestry of the central metabolism of *T. ammonificans*. Our genome reconstruction and phylogenetic analyses indicate that the core or ancestral groups of genes involved in the central metabolism of *T. ammonificans* include genes coding for Group 3 hydrogenases, genes coding for enzymes of the reductive Acetyl-CoA pathway and genes involved in sulfur reduction. These pathways are catalyzed by enzymes that are extremely sensitive to oxygen which are either present in early branching Archaea, or are directly involved in metabolic reactions that do not require oxygen (or oxygen by-products) and use substrates of geothermal origin. Weiss et al. reconstruction of the LUCA genome is surprisingly consistent with our findings: the authors conclude that “FeS and transition metals are relics of ancient metabolism, that life arose at hydrothermal vents and that sulfur was involved in ancient metabolism”. Similarly to *T. ammonificans*, LUCA was an H2-dependent autotroph capable of S utilization that could fix CO2 via the reductive Acetyl-CoA pathway. What this means is that part of the genome of *T. ammonificans*, a modern bacterium that lives today at deep-sea vents, bears some striking similarities to the putative genome of LUCA as reported by Weiss et al. Therefore, part of the central metabolism of *T. ammonificans* is ancestral. We added a sentence to the third paragraph of the subsection “Comparative genomic of *T. ammonificans*: ancestral and acquired metabolic traits” with a reference to the Weiss et al. paper.

*Subsection “Comparative genomic of T. ammonificans: ancestral and acquired metabolic traits”, last two paragraphs: This is an interesting idea, but where is your data supporting this hypothesis? No figures or tables are cited in the text.*

Comment on the oxygen radical detoxification enzymes being acquired as a response to oxygen exposure. This is a hypothesis generated on the widely-accepted assumption that prior to the onset of oxygenic photosynthesis the Earth was anoxic. Hence, it is reasonable to posit that the oxygen radical detoxification machinery is a relatively “late” acquisition in the lineage that led to *T. ammonificans*. We are currently carrying out experiments to test the function of these enzymes in *T. ammonificans*. Such experiments go beyond the scope of this paper.

*Subsection “Insight into the evolution of carbon fixation”, first paragraph: The Brakeman and Smith study is outdated, more updated is Weiss et al. 2016 Nature Microbiology.*

Comment on the Braakman and Smith study being “outdated”, and the underlying suggestion by reviewer 2 to refer exclusively to the Weiss et al. study. We disagree. First, the Weiss et al. paper was published after our manuscript was submitted to *eLife*. Second, the Weiss et al. paper does not refute the Braakman and Smith findings. Rather, despite the radically different approaches used in the two studies – the Braakman and Smith is based on phylometabolic reconstruction while the Weiss et al. is based on gene phyletic pattern reconstruction – both point to the basal lineage being autotrophic and using the reductive Acetyl-CoA pathway. We included a reference to the Weiss paper in the first paragraph of the subsection “Insight into the evolution of carbon fixation”.

*Subsection “Insight into the evolution of carbon fixation”, end of first paragraph: I disagree, new analyses of all prokaryotic genomes and biochemical pathways looking at the oldest conserved genes between all organisms suggest LUCA was a methanogen with reductive acetyl CoA pathway (Weiss et al. 2016). Not Aquificae as the authors propose here.*

Reviewer 2 disagrees with the conclusion of the Braakman and Smith study, states that Weiss et al. demonstrate that LUCA was a methanogen, and implies that we propose that LUCA was an *Aquificae*. First, Weiss at al. suggest that LUCA’s genomic makeup points to an autotroph that fixed carbon via the reductive acetyl-CoA pathway, and suggest acetogenic and/or methanogenic roots. They do not unequivocally show that LUCA was a methanogen. Second, we are *not* proposing that LUCA was an *Aquificae*. Instead, we are reporting findings from Braakman and Smith, which are partly in agreement with Weiss et al. (please see the paragraph above), and we discuss that our finding of the reductive acetyl-CoA pathway in *T. ammonificans* is in line with the hypothesis that this pathway of autotrophic carbon fixation was present in the basal lineage. Again, this is in agreement with findings from BOTH Braakman and Smith and Weiss et al.

*Reviewer #3:*

*This is a very comprehensive and beautifully written genomic & proteomic analysis of Thermovibrio ammonificans, followed by a comparative analysis of its C fixation pathways with those of the Aquificaceae and the sulfur/hydrogen-oxidizing epsilonproteobacteria. On first reading, I can only find minor details that need attention (see below). After seeing how cursory many genomes are treated, it was a pleasure to read this analysis and its careful exploration of autotrophic pathway evolution.*

*The underpinnings of this manuscript – dividing biochemical pathways into ancestral and modern, and then dissecting the genome of a bacterium into ancient and modern layers, like different time horizons in an archaeological excavation – require a short comment. The complexity of even "ancient" pathways, genes and enzymes reflects the fact that every "ancient" gene and enzyme that still exists today has a billion-year history of evolutionary finetuning and, often enough, lateral gene transfer and functional repurposing. Nothing has survived from the dawn of life unchanged. This short preface should explain why I take some introductory statements of this article with a grain of salt, for example the big jump from Wachterhauser to the rTCA cycle (see below).*

*Introduction, second paragraph: If this assessment – the Aquificae as an early lineage – is based on 16S rRNA gene sequences, you should say this explicitly.*

Introduction, second paragraph: The assessment that *Aquificae* are an early-branching lineage is based on phylogenetic analyses of both the 16S rRNA gene and whole genomes. We added a sentence stating that.

*Introduction, third paragraph: I don't have access to Waechtershauser 1988 right now, but his article is focusing on surface catalysis of CO_2_, CO and small organic molecules being assembled into larger molecules, all on a metal-sulfide or pyrite surface; the theme is prebiotic "autotrophic" syntheses catalyzed by a reactive surface. The fully developed rTCA cycle is far away from that.*

The comment about Wächtershäuser’s iron-sulfur-based prebiotic synthesis model being far away from a functional rTCA cycle is correct. To avoid ambiguities, we removed the references to Wächtershäuser’s work associated to this specific statement. However, the idea that Wächtershäuser’s proposed autocatalytic “surface” metabolism eventually transitioned to a primitive, sulfur dependent version of modern rTCA cycle is embedded in his work and is presented further down the paragraph (Introduction, third paragraph).

*The last sentence of the subsection “Conclusions” – "we argue that proposed model for the evolution of metabolism presented in this paper may represent the blueprint of the chemistry of early life" – goes too far in the same direction. Most problematically in origin of life scenarios, early probiotic reactions, like covalent bond formations at reactive interfaces or charge separation at mineral surfaces, are separated by a huge gulf from fully developed enzymatic systems, encoded by genes, transcribed by mRNA, translated by ribosomes… you get the idea. Even ancestral enzymes and pathways only provide a view on the evolution of fully differentiated, metabolically active and self-sustaining microorganisms that we would recognize today [with proper genomes, not to forget], but they do not allow a view to the really early stages and origin of life; theories on communal genomes before cell differentiation, and early evolution of protein domains and structural subunits try to fill the gap. This difference should be acknowledged. Again, nothing has survived from the dawn of life unchanged.*

We removed the reference to the chemistry of early life from the concluding sentence and emphasized the importance of using early branching microorganisms to investigate the evolution of metabolism (subsection “Conclusions”).

[Editors' note: further revisions were requested prior to acceptance, as described below.]

*Following are the key issues that need to be addressed:*

*Much of the rebuttal consists of straw man arguments regarding the Weiss et al. study and origins of life issue that detract from my main points raised about their analyses. I apologize if the authors got the wrong idea from my original review, which was not to ask them to re-write their manuscript on the origins of life. I was not a co-author on that paper, but it is a good example of extensive taxon sampling and (in principle) a good method for determining vertical inheritance. As the authors note in their rebuttal whether or not Weiss et al. actually followed their method is currently debated in online comment forums, but this is outside the scope of the authors’ manuscript. By focusing on this, I think that the authors may have missed the main point of my review, which was that their methods for phylogenomic analyses were not sufficient (or at least not sufficiently explained in the Methods). It is possible that analyses were done satisfactorily but this is impossible to tell from the information provided. I have re-read the authors Methods carefully a second time in the revised manuscript, yet it is not possible to tell how the authors processed their data in the different analyses.*

*My main point is that the authors need to explain in their Methods exactly how they did the different phylogenetic analyses, and provide a satisfactory justification for why they chose to treat their data in the way they did (e.g., why no concatenated gene phylogenies?) instead of following other conventional methods for interpreting bacterial phylogenomics. For example, concatenated gene trees possess more phylogenetic signals, and are less susceptible to the stochastic errors than those built from single genes (Wu and Eisen, 2008; Jeffroy et al. 2006). From what I can tell the authors performed single gene phylogenies of a handful of genes picked from the genomes.*

In this second revision, we have significantly amended our Methods section (see the Methods section in the revised manuscript) and taken further step to ensure reproducibility of our analyses. We have also expanded the phylogenetic section by adding additional information requested by reviewer 2, including the number of sites used for the reconstruction. We strongly believe that reproducibility in science is a very important issue, and we strive to make our methods and data publicly available. All comparative genomic analysis and alignments carried out were already directly accessible as permanent url in the public genomeevolution.org server.

In order to identify the ancestral and acquired portion of the central metabolism of *T. ammonificans*, we used a phylometabolic – rather than a phylogenomic – approach. In phylometabolic analyses, the metabolic pathways of the organism under investigation are compared to those found in related organisms both within and across neighboring clades. By focusing on the pathways, the comparison may reveal variations in multi-enzyme functional units, providing context for the completion of the pathway within the networks of individual organism, while also allowing for the identification of ancestral states and horizontal gene transfer events. The resulting phylometabolic tree includes multiple complete pathways to common essential metabolites, and suggests which evolutionary substitutions are allowed (at either organism or ecosystem levels) among these pathways (see Braakman and Smith, 2012 and reference therein for a more extensive description of the principles underlying this approach and more recently Braakman et al., 2017 for ecological application of this principles). In this second revision we included a section in the Methods explaining the rationale underlying phylometabolic approaches.

*The authors write in their rebuttal: "That is incorrect, as members of the Desulfurobacteriaceae, Aquificaceae and Epsilonproteobacteria were included in the analyses and discussed throughout the manuscript. We used all available Aquificae genomes (as indicated in Table 3), and further focused only on the closest relative and the Epsilonproteobacteria as an ecological proxy."*

*A quick search of all available genomes from the Aquificae in IMG reveals that there are genomes from 29 representatives available for analysis. Table 3 in the manuscript contains 17 of these, which cover the major clades but does not include all available genomes from each clade. Thus, the statement that the authors make that "Comparative analyses between the genomes of T. ammonificans, all available Aquificae genomes…" is incorrect. Taxon sampling (e.g., n=17 vs. n=29) can have an impact on the results of phylogenetic analyses. Usually the addition of more representatives can provide a better resolution. Please justify why you selected only these 17 for analysis.*

*In the Methods section "Comparative genomics": It is clear from this that you used the genomes in Table 3. But, it is not clear whether these same genomes were used to create the phylogenetic analyses or not.*

At the time of writing, a quick search on the NCBI genome page will confirm the availability of 23 genomes divided in 17 *Aquificae* species. However, the number of available genomes is constantly increasing (the genome of *Thermocrinis minervae* was released in 2016/12/02 while *Desulfurobacterium indicum* was released in 2017/01/23; please see the list at the end of this letter).

In our analysis we included all the available genomes of validly published *Aquificae* species – that is, species that have been physiologically characterized and are recognized by the International Committee on Systematics of Prokaryotes – and for which the type strain is available from a cell culture collection (e.g., ACCT or DSMZ). This will allow for future studies based on our findings to go beyond comparative genomics and test our hypotheses experimentally, using comparative physiological, transcriptomic and proteomic analyses. However, in order to maximize the diversity of our genome dataset, we made two exceptions to this rule. The first exception was to include in the analysis one genome to represent the genus *Hydrogenivirga* and one to represent the genus *Hydrogenobaculum* (for which we selected the current reference genome, see NCBI page), as no genome sequence is available for validly published species of these genera. The second exception was to select two additional genomes belonging to the genera *Desulfurobacterium* (the closest relative to the genus *Thermovibrio*) and *Persephonella,* respectively.

The genomes excluded from our analysis are:

Four closely related genomes of the genus *Hydrogenobaculum* (average gapped genome similarity above 94% to the included *Hydrogenobaculum* sp. Y04AAS1 genomehttps://www.ncbi.nlm.nih.gov/genome/neighbors/13671?genome_assembly_id=301047)

An alternative assembly of *Hydrogenobacter thermophilus* TK-6 already included in the analysis

One genome of the genus *Persephonella* (gapped genome similarity above 92% to *Persephonella* sp. IF05-L8 included in the analysis,https://www.ncbi.nlm.nih.gov/genome/neighbors/32114?genome_assembly_id=202711).

*In the Methods section "Phylogenetic Methods": The authors describe methods for 16S phylogenies and ATP citrate lyase phylogenies. No information is provided regarding the Ni-Fe hydrogenase phylogeny (Figure 5), CO dehydrogenases (Figure 4), or any other phylogenetic analyses that the authors base their conclusions on. At least, not from what I can tell after reading the section. I do not know whether this is intended or a mistake. Since the authors base their conclusions based on these results it seems like a good idea to explain how these analyses were done. For example, please state number of aligned resides for each tree, whether it is amino acids of DNA data, e.g. the standard information that readers deserve to know so that the authors work can be openly evaluated by the scientific community after publication. Another major question remains, which is why have the authors performed only single gene phylogenies as opposed to concatenated gene phylogenies? Please explain. Concatenated gene phylogenies are less susceptible to stochastic errors than those built from single genes.*

We agree with reviewer 2 about the usefulness of concatenated phylogenies. However, the use of concatenated vs. single protein phylogenies depends on the objective of the study. Concatenated phylogenies are a powerful tool to resolve the phylogenetic relationships among organisms, and undoubtedly are less susceptible to stochastic errors (Wu and Eisen, 2008). Concatenated phylogenies are thus the appropriate mode of analysis when the objective is to resolve organismal relationships. However, concatenated phylogenies cannot be used to reconstruct the phylogenetic history of a gene (unless the concatenated phylogeny is performed *exclusively* on the multiple subunit of the same protein complex, as we have performed in the present study for the ATP-citrate lyase). Phylogenies of single enzyme (that can include multiple subunits) are necessary to reconstruct the phylogenetic history of the metabolic pathways they represent. In this context, the use of concatenated phylogenies, which include different genes or enzymes, can mask lateral gene transfer events, which need to be investigated singularly (Boussau et al., 2008; Philippe and Douady, 2003; Bapteste et al., 2004). Our objective was to identify the ancestral and acquired complement of the genome of *T. ammonificans*, and compare it with the genomes of other members of the *Aquificae*. A revision of the overall phylogenetic relationship among the different species and families of the *Aquificae* was not the object of this study. Rather, we aimed at reconstructing the history of key enzymes diagnostic of specific biochemical pathways, and at comparing the reconstructed histories of such key enzymes among members of *Aquificae*.

Therefore, we believe that the use of single protein phylogenies is appropriate to the objectives of our study.